# Tissue-resident natural killer cells support survival in pancreatic cancer through promotion of cDC1-CD8 T activity

**Simei Go**[1†], **Constantinos Demetriou**[1†], **Giampiero Valenzano**[1†], **Sophie Hughes**[1], **Simone Lanfredini**[1], **Helen Ferry**[2], **Edward Arbe-Barnes**[3], **Shivan Sivakumar**[1], **Rachel Bashford-Rogers**[4], **Mark R Middleton**[1,2,5], **Somnath Mukherjee**[5], **Jennifer Morton**[6,7], **Keaton Jones**[8]*, **Eric O Neill**[1]*

[1]Department of Oncology, University of Oxford, Oxford, United Kingdom; [2]Experimental Medicine Division, University of Oxford, Oxford, United Kingdom; [3]University of Oxford Medical School, Oxford, United Kingdom; [4]Department of Biochemistry, University of Oxford, Oxford, United Kingdom; [5]Oxford University Hospitals NHS Foundation Trust, Oxford, United Kingdom; [6]CRUK Beatson Institute, Glasgow, United Kingdom; [7]School of Cancer Sciences, University of Glasgow, Glasgow, United Kingdom; [8]Nuffield Department of Surgical Sciences, University of Oxford, Oxford, United Kingdom

*For correspondence:
keaton.Jones@nds.ox.ac.uk (KJ);
eric.oneill@oncology.ox.ac.uk
(EON)

†These authors contributed
equally to this work

**Competing interest:** The authors
declare that no competing
interests exist.

**Reviewing Editor:** Robert
Baiocchi, The Ohio State
University, United States

## eLife assessment

This **valuable** manuscript provides an interesting account documenting the role of resident CD56(br) NK cells in driving interaction with dendritic cells that attract CD8+ T cells to the pancreas cancer tumor microenvironment (TME). The work **convincingly** illustrates how irradiation combined with CCR5i and PD1 blockade leads to a reduction in pancreatic cancer growth that correlates with a reduction in Treg cells and enhancement of NK and CD8 T cells in the TME. The correlation of NKC1 signature with survival in pancreatic cancer patients is indeed of broader interest regarding potential relevance to other types of cancer.

**Abstract** The immunosuppressive microenvironment in pancreatic ductal adenocarcinoma (PDAC) prevents tumor control and strategies to restore anti-cancer immunity (i.e. by increasing CD8 T-cell activity) have had limited success. Here, we demonstrate how inducing localized physical damage using ionizing radiation (IR) unmasks the benefit of immunotherapy by increasing tissue-resident natural killer (trNK) cells that support CD8 T activity. Our data confirms that targeting mouse orthotopic PDAC tumors with IR together with CCR5 inhibition and PD1 blockade reduces E-cadherin positive tumor cells by recruiting a hypoactive NKG2D$^{-ve}$ NK population, phenotypically reminiscent of trNK cells, that supports CD8 T-cell involvement. We show an equivalent population in human single-cell RNA sequencing (scRNA-seq) PDAC cohorts that represents immunomodulatory trNK cells that could similarly support CD8 T-cell levels in a cDC1-dependent manner. Importantly, a trNK signature associates with survival in PDAC and other solid malignancies revealing a potential beneficial role for trNK in improving adaptive anti-tumor responses and supporting CCR5 inhibitor (CCR5i)/αPD1 and IR-induced damage as a novel therapeutic approach.

## Introduction

Over the past decade, immune checkpoint inhibitors have shown significant success in the treatment of various solid malignancies. Treatment of pancreatic cancer using immunotherapy (IT) alone or in combination with radiotherapy or chemotherapy has unfortunately seen limited clinical success over either modality alone (*Bockorny et al., 2020*; *Doi et al., 2019*; *Royal et al., 2010*; *Weiss et al., 2018*; *Weiss et al., 2017*; *Mohindra et al., 2015*). This is likely due to several tumor microenvironmental factors, including the dense stroma that supports an immunosuppressive environment while also obstructing the infiltration of cytotoxic immune cells (*Mills et al., 2022*; *Piper et al., 2023*). Novel strategies comprising dual targeting of (programmed cell death-1) PD-1/IL-2Rβγ with radiotherapy are beginning to indicate that potential benefits may require a coordinated alteration in the suppressive microenvironment involving loss of regulatory T-cells (Tregs) and increased natural killer (NK) cell infiltration, in addition to simply increasing CD8 T-cell infiltration (*Piper et al., 2023*).

We previously identified serum CCL5 (RANTES) as a negative prognostic marker for late-stage advanced pancreatic cancer (*Willenbrock et al., 2021*). CCL5 is pro-inflammatory and an acute response to injury or infection that promotes recruitment of monocytes and lymphoid immune cells through the receptors CCR5, CCR3, and CCR1. In cancer, sustained CCL5-mediated inflammation leads to a suppressive environment via attraction of CCR5$^+$ immunoregulatory components including Tregs, tumor-associated macrophages (TAMs), and myeloid-derived suppressor cells (MDSCs) normally involved in resolving inflammatory events (*Hemmatazad and Berger, 2021*). Pancreatic tumors that are poorly differentiated produce higher levels of CCL5 and express more of its main receptor, CCR5, compared to well-differentiated and non-cancerous tissue (*Monti et al., 2004*; *Singh et al., 2018*). Disrupting this loop by the FDA-approved CCR5 inhibitor (CCR5i) maraviroc reduces pancreatic tumor cell migration, invasiveness (*Singh et al., 2018*), and proliferation (*Huang et al., 2020*). Silencing CCL5 expression in Panc02 cells or systemic administration of the CCR5i TAK-779 reduces Treg migration and tumor volume in a murine subcutaneous tumor model (*Tan et al., 2009*). These studies suggest that the CCL5/CCR5 axis may be a promising therapeutic target in pancreatic cancer as it has the potential to alter both the intrinsic properties of tumor cells and immune cell migration.

In established pancreatic tumors, the immune suppressive environment promotes more tissue repair/resolution signaling and prevents pro-inflammatory signaling (*Mantovani et al., 2017*). Radiotherapy is delivered to localized tumors to stimulate a pro-inflammatory environment and increase the opportunity for tumor neoantigens to be recognized by infiltrating adaptive response cells (*Benkhaled et al., 2023*). As such, stereotactic ablative radiotherapy delivers high doses of radiotherapy in a minimal number of fractions to maximize an inflammatory cascade but the sensitivity of lymphocytes to DNA damage and the immune suppressive environment prevent benefit or meaningful tumor control (*Mills et al., 2022*). In NSCLC, single-cell RNA sequencing (scRNA-seq) of immune cells within tumors identifies a distinct subset of CD49a$^+$ CD103$^+$ tissue-resident memory (TRM) CD8$^+$ T-cells that have capacity to respond to neoantigens but are functionally suppressed (*Caushi et al., 2021*). In this case, the use of an anti-PD1 antibody (αPD1) supports TRM activation but additional disruption of the immune suppressive environment is still required, indicating that additional components are required in addition to simply activating CD8$^+$ T-cell neoantigen recognition.

Kirchhammer et al. recently reported that a tissue-resident NK (trNK) cell population is induced in NSCLC in response to viral delivery of IL-12, which crucially supported type 1 conventional dendritic cell (cDC1) infiltration and increased DC-CD8$^+$ T-cell interactions (*Kirchhammer et al., 2022*). Together with PD1 blockade, IL-12-mediated recruitment of trNK cells enhances cross-presentation of antigen to CD8 T via cDC1, suggesting this represents a substantial barrier for T-cell focused therapy and that improving the NK/DC/T-cell crosstalk can promote anti-tumor immunity and tumor control. Notably, this was dependent on CCL5 and points to a positive role for inflammatory CCL5 signaling in the absence of CCR5-mediated suppression (*Kirchhammer et al., 2022*).

Given the negative prognosis associated with high CCL5 serum levels in human patients and its diverse functions in chemokine and paracrine signaling (*Aldinucci et al., 2020*), we hypothesized that inhibiting CCR5 rather than CCL5, in combination with radiotherapy and anti-PD1 IT, may improve tumor control via a multimodal approach. To test this, we employed a murine orthotopic pancreatic cancer model and monitored for tumor growth and immune infiltration. We find that while the use of a CCR5i alone restricts Treg involvement, it did not therapeutically beneficially impact tumor viability alone or together with αPD1. However, in combination with radiotherapy, we see a significant

alteration in MDSCs, NK, and CD8 T-cells and better tumor control. Interestingly, we observe that IR/CCR5i/αPD1 combination treatment induced a trNK population in which the NKG2D- NK cells are the highly correlated immune population with tumor control. Exploration of scRNA-seq datasets from human pancreatic ductal adenocarcinoma (PDAC) studies confirms the presence of trNK cells as immunomodulatory in human PDAC, and in which these cells seem to directly support cDC1-CD8 communication. Strikingly, a specific trNK signature indicates that higher levels of this NK subtype are significantly correlated with better PDAC patient overall and disease-free survival. Moreover, pan-cancer analysis reveals that trNK cell involvement is associated with better patient survival across a number of solid tumors and supports the potential utility of a combination regimen comprising ionizing radiation (IR) and CCR5i/αPD1 IT as a promising strategy to increase NK/cDC1/CD8-mediated tumor control in solid cancers.

## Results

### CCL5 is a negative prognostic marker in pancreatic cancer

We previously identified serum CCL5 as a bona fide negative prognostic marker for pancreatic cancer (*Willenbrock et al., 2021*, and found that, in two independent PDAC cohorts, CPTAC3 [*Cao et al., 2021*] and TCGA [*Uhlen et al., 2017*]), higher CCL5 expression associates with poor overall and disease-free survival, confirming the negative implication of elevated levels of CCL5 in pancreatic cancer (*Figure 1A*). To understand the cause of CCL5-mediated reduced survival in PDAC, we hypothesized that immune cells responsive to a CCL5 chemotactic gradient (through expression of the cognate receptors CCR5, CCR3, or CCR1) could be potential contributors to an adverse tumor immune environment (*Figure 1B*, *Figure 1—figure supplement 1A*). scRNA-seq of human tumors has revealed genetic signatures for myeloid-derived suppressor cells of polymorphonuclear (PMN-MDSC) or monocytic origin (*Alshetaiwi et al., 2020*), TAMs (*Wang et al., 2021*), CD4 Tregs (*Mijnheer et al., 2021*), and NK cells (*Smith et al., 2020*), but none of these immune populations have yet been implicated as a causative agent in the poor outcome associated with high CCL5 expression in PDAC (*Figure 1C*). Notably, exploration of individual genes within these signatures indicated stark opposing correlations within each signature pool - particularly for genes associated with MDSCs (*S100P* vs *ARG2*) and NK cells (CD56 vs CD16) (*Figure 1C*, *Figure 1—figure supplement 1B*). CD56 (*NCAM1*) is a key marker that represents functionally distinct subpopulations of NK cells where CD56$^{bright}$ NK cells represent immature states that move to CD56$^{dim}$ upon maturation to full cytotoxic potential, whereas conversely CD16$^+$ expression marks activation and CD16$^-$ immature or quiescent NK cells (*Poli et al., 2009*). Thus, these distinct subtypes may differentially contribute to survival, e.g., association of CD16 with poor survival may implicate a detrimental role of CD56$^{dim}$CD16$^+$ NK cells, whereas the strong positive prognosis associated with CD56 expression could indicate benefit of CD56$^{bright}$CD16$^-$ NK cells. The benefit of CD56 can be attributed to NK cells over neuronal expression as the neuronal cell-specific homolog *NCAM2* has no prognostic value (*Figure 1—figure supplement 1C*).

### CCR5i modulates Treg infiltration in a murine orthotopic pancreatic tumor model

We next employed a syngeneic orthotopic pancreatic cancer model where cells derived from *Kras$^{G12D}$*, *Trp53$^{R172H}$*, *Pdx1-Cre* (KPC) tumor bearing mice are injected into the pancreas of wildtype C57BL/6 mice, thereby recapitulating a human pancreatic cancer microenvironment (*Matzke-Ogi et al., 2016*). From a selection of independently isolated KPC-derived cell lines, we first determined the cell line KPC-F as an appropriate model for human PDAC tumor based on the expression of epithelial (E-cadherin), mesenchymal (vimentin), and stromal (Collagen I, αSMA) markers as well as displaying growth kinetics (determined by MRI) amenable for the study and expression of high levels of CCL5 (*Figure 2A*, *Figure 2—figure supplement 1A and B*). To address which immune subsets are involved in CCL5 signaling in pancreatic cancer, we developed a 17-color spectral flow cytometry panel to monitor the tumor-infiltrating immune microenvironment in parallel (*Figure 2—figure supplement 1C*) and employed a specific CCR5i (maraviroc), to block CCR5, the most widespread and highest-affinity receptor for CCL5 expressed on Tregs (*Figure 1—figure supplement 1A*).

Pilot experiments injecting 500 KPC-F cells yielded robust and replicative growth kinetics (indicated by matching volumes of ±100 mm$^3$ at the beginning of exponential growth), permitting direct

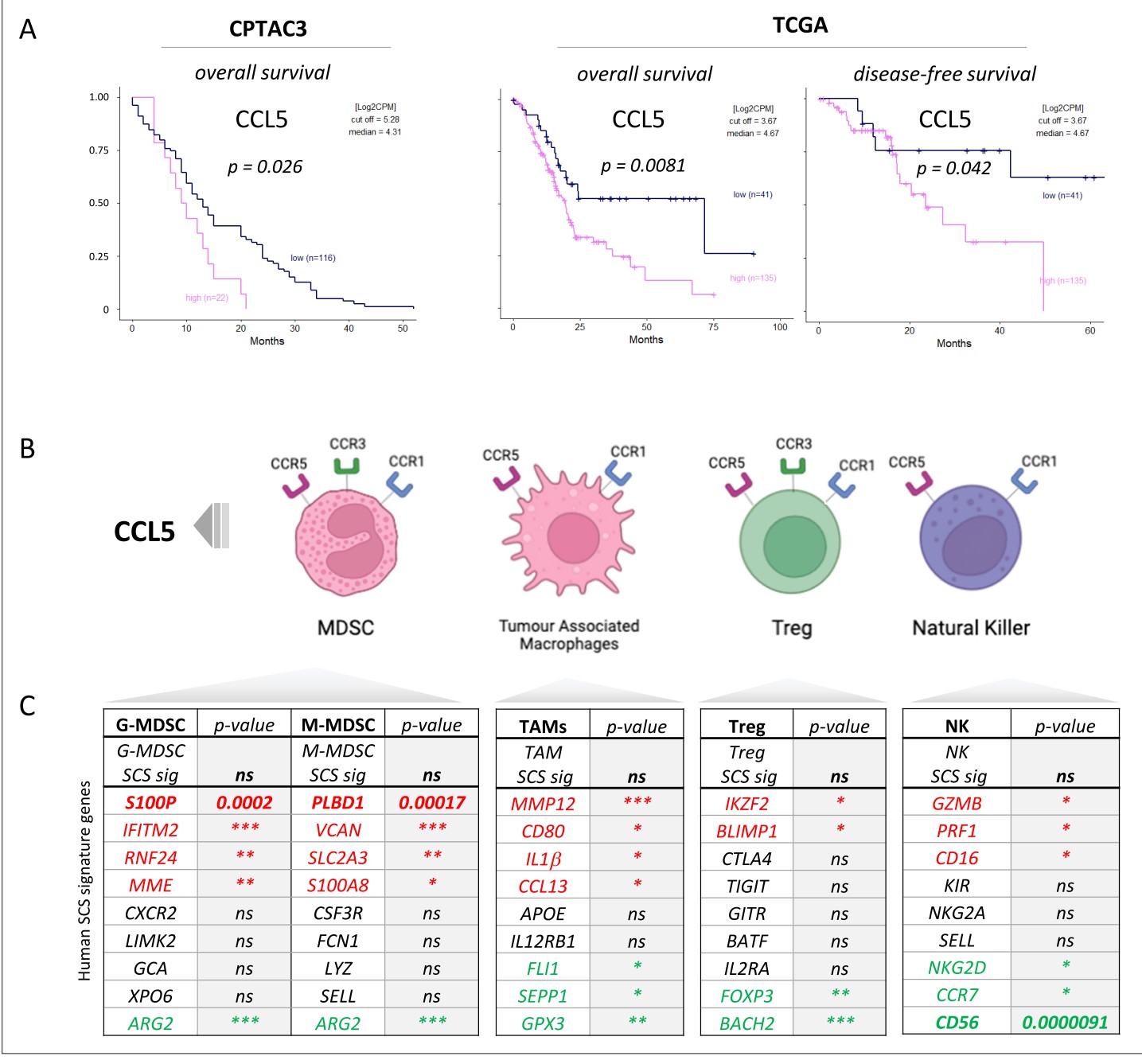

**Figure 1.** CCL5 is clinically significant in pancreatic cancer. (**A**) Overall and disease-free Kaplan-Meier survival plots of pancreatic ductal adenocarcinoma (PDAC) patients segregated into high or low CCL5 gene expression levels within pancreatic tumors. Data are derived from CTPAC3 and TCGA cohorts and optimal cut-off values were calculated using the max-stat method for each respective cohort. (**B**) Schematic overview of CCL5-responsive immune cells and corresponding CCL5 receptor repertoire expression. (**C**) Correlation between overall and single-cell gene signatures of CCL5-responsive immune cells with overall PDAC prognosis. Color depicts positive (green), negative (red), or neutral (black) prognostic outcomes (*p<0.05, **p<0.01, ***p<0.005). Data are derived from the Pathology Dataset of the Human Protein Atlas and based on human tissue micro arrays and correlated log-rank p-value for Kaplan-Meier analysis αPD1/CCR5i/IR (n=8).

The online version of this article includes the following figure supplement(s) for figure 1:

**Figure supplement 1.** CCL5 receptors are distributed across different immune cells.

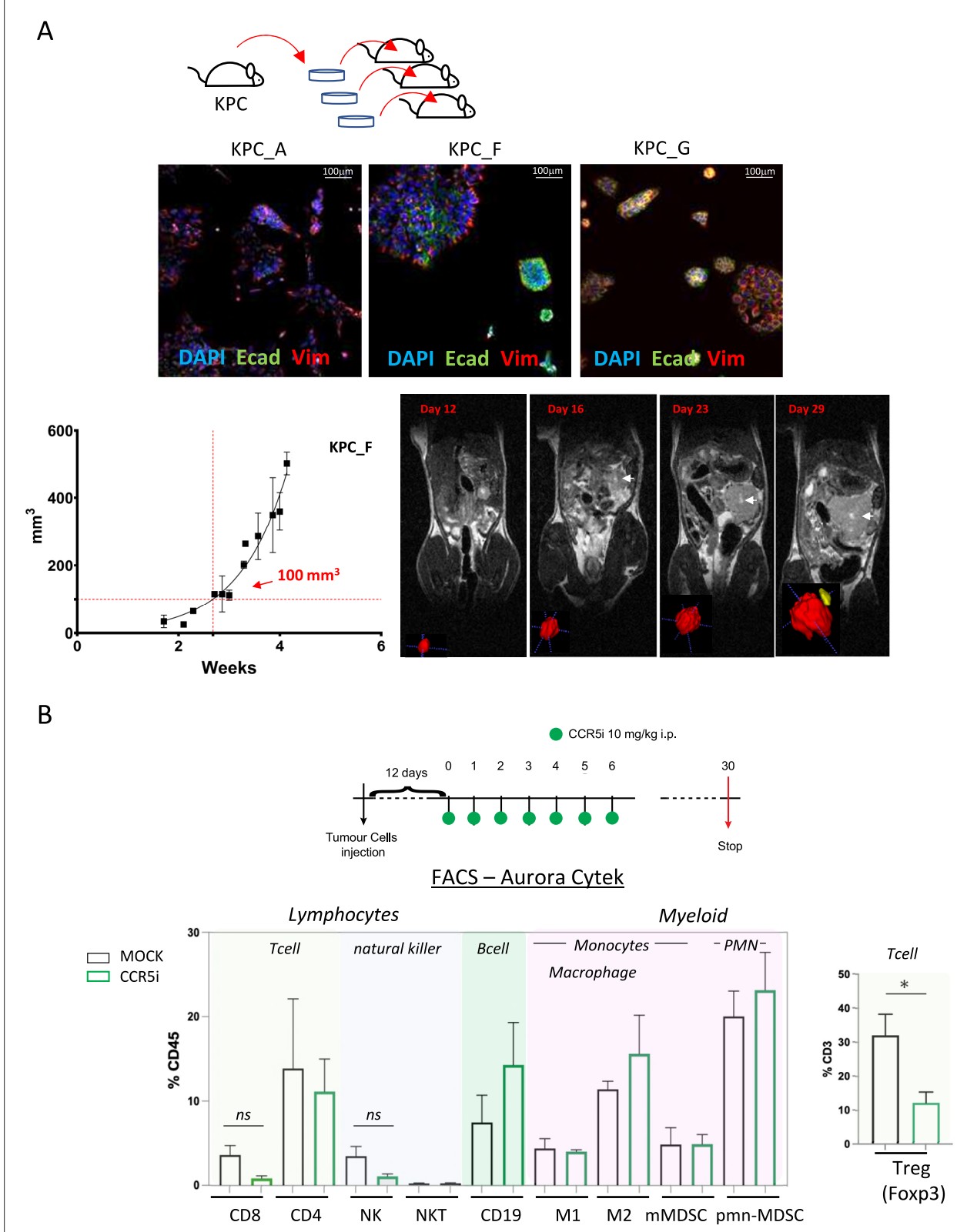

**Figure 2.** CCR5i restricts Tregs in a pancreatic ductal adenocarcinoma (PDAC) orthotopic tumor model. (**A**) Three different lineages of KPC pancreatic tumor cells (derived from Kras$^{G12D}$p53$^{R172H}$Pdx1-Cre mice) were obtained and stained for DAPI (blue; nucleus), E-cadherin (green; epithelial), and vimentin (red; mesenchymal). Growth curve of orthotopically injected KPC-F cells (500cells) into the pancreas of wildtype C57BL/6 over time in weeks. Tumor volume was measured using MRI. Representative MRI images over time are displayed, white arrow denotes tumor mass. (**B**) Timeline of maraviroc (CCR5

*Figure 2 continued on next page*

*Figure 2 continued*

inhibitor [CCR5i]) treatment regimen. A total of 12 days post-orthotopic injection, tumor-bearing mice were treated daily with maraviroc (10 mg/kg via intraperitoneal injection) for 6 days and followed for up to 30 days after starting treatment. Frequencies of pancreatic tumor-infiltrating immune cells harvested at day 30 with or without maraviroc using spectral flow cytometry are shown. Data are represented as mean percentage positive cells of Live/CD45$^+$ cells ± SD. For Tregs, the mean percentage positive cells of Live/CD45$^+$ CD3$^+$ cells ± SD are shown. Significance was tested using the Welch and Brown-Forsythe ANOVA for parametric data or Kruskal-Wallis test for non-parametric data. Mock (n=6), IR (n=3), aPD1 (n=8), aPD1+IR (n=8), CCR5i (n=3), CCR5i+IR (n=8), aPD1+CCR5i (n=5), αPD1/CCR5i/IR (n=8). *p<0.05.

The online version of this article includes the following figure supplement(s) for figure 2:

**Figure supplement 1.** Set up for the mouse orthotopic model immune monitoring.

comparisons of infiltrating immune cells across different treatment groups (*Figure 2A*). Next, tumors were allowed to reach 50 mm$^3$ (approx. 12 days) before initiation of treatment with daily administration of CCR5i for 7 days. Mice were culled 30 days post-treatment for the characterization of infiltrating immune cells (*Figure 2B*). Mice in the CCR5i-treated group showed a trend toward reduction in NK cells and all T-cell subsets with a significant reduction in Treg (Live/CD45$^+$CD3$^+$CD4$^+$FOXP3$^+$) infiltration, indicating the active recruitment of CCR5$^+$ Tregs in PDAC (*Figure 2B*). Given the enhanced effect of CCR5i on tumor growth kinetics (*Figure 3A*), however, standalone inhibition of CCR5 is unlikely to clinically benefit PDAC progression.

## Combination immunotherapy following localized damage alters tumor immune composition

Induction of a localized inflammatory microenvironment with IR can drive anti-tumor responses, but there is limited evidence for clinical benefit of radiotherapy in combination with immune checkpoint inhibitors in PDAC (ClinicalTrials.gov identifier: NCT04098432) (*Chen et al., 2022*). Therefore, we next examined whether the addition of CCR5i to tumor-targeted IR could improve this by further modulating immune cell migration, in particular by reducing Treg involvement (*Figure 3A*, *Figure 3—figure supplement 1A*). As expected, radiotherapy (3×4 Gy) produced a strong effect on gross tumor volume, significantly reducing volumes over standalone treatment with αPD1, CCR5i, or CCR5i+αPD1 (*Figure 3A*). Responses to combination of IR with CCR5i, αPD1, or CCR5i+αPD1 (IR+IT) showed larger variations compared to IR alone, implying mixtures of 'responders' and 'non-responders' to immune-modulatory treatments. Moreover, IR/CCR5i/αPD1-treated tumor sections had significantly reduced DAPI$^+$ and p53$^+$ KPC cells compared to all other conditions, suggesting significantly more loss of tumor cells by triple combination treatment (*Figure 3B*, *Figure 3—figure supplement 1B and C*). Tumor sections from the triple combination treatment also presented with increased loss of active stroma (αSMA staining, *Figure 3—figure supplement 1D*) and increased necrotic areas over standalone radiotherapy (*Figure 3C*). In line with apparent increased tumor control, the triple combination treatment demonstrated infiltration of CD8$^+$ T-cells (*Figure 3D*), supporting greater penetration of CD8$^+$ T-cells into the center of tumors (*Figure 3—figure supplement 1E*). These results support improved tumor control with combination of IR/CCR5i/αPD1 over standalone radiotherapy or IR in combination with αPD1 and CCR5i alone. To elucidate the immune mechanisms behind this control, we analyzed the immune infiltrate of pancreatic tumors. In line with the results above, the triple combination reduced Treg infiltration, enhanced CD8$^+$ T-cell infiltration, and supported a moderate increase in CD4$^+$ T helper cells (*Figure 3E*, *Figure 3—figure supplement 1F*).

Interestingly, no alteration was seen in the myeloid compartment, except for a reduced infiltration of PMN-MDSCs, whereas a significant infiltration of NK and NKT cells could be observed in pairwise comparisons between IR+αPD1 vs IR+CCR5i or IR/CCR5i/αPD1 (*Figure 3E*, *Figure 3—figure supplement 1F*). These results collectively support the notion that IR/CCR5i/αPD1 combination treatment alters immune infiltration by reducing Tregs and increasing NK and CD8 T-cells, thereby resulting in greater local tumor control.

## CD8 T- and NKG2D$^-$ NK cells correlate with increased tumor control

To derive more granularity on the potential roles of NK cell and T-cell populations in tumor control, tissue sections were stained using an optimized multiplex immunofluorescence panel and analyzed by HALO AI software to identify tumor cells (E-cadherin$^+$ - blue), CD8$^+$ T-cells (CD3$^+$CD8$^+$ - orange), CD4$^+$ T-cells (CD3$^+$CD8$^-$ - green), and NK cells (CD3$^-$CD8$^-$NK1.1$^+$ - red) (*Figure 4A*, *Figure 4—figure*

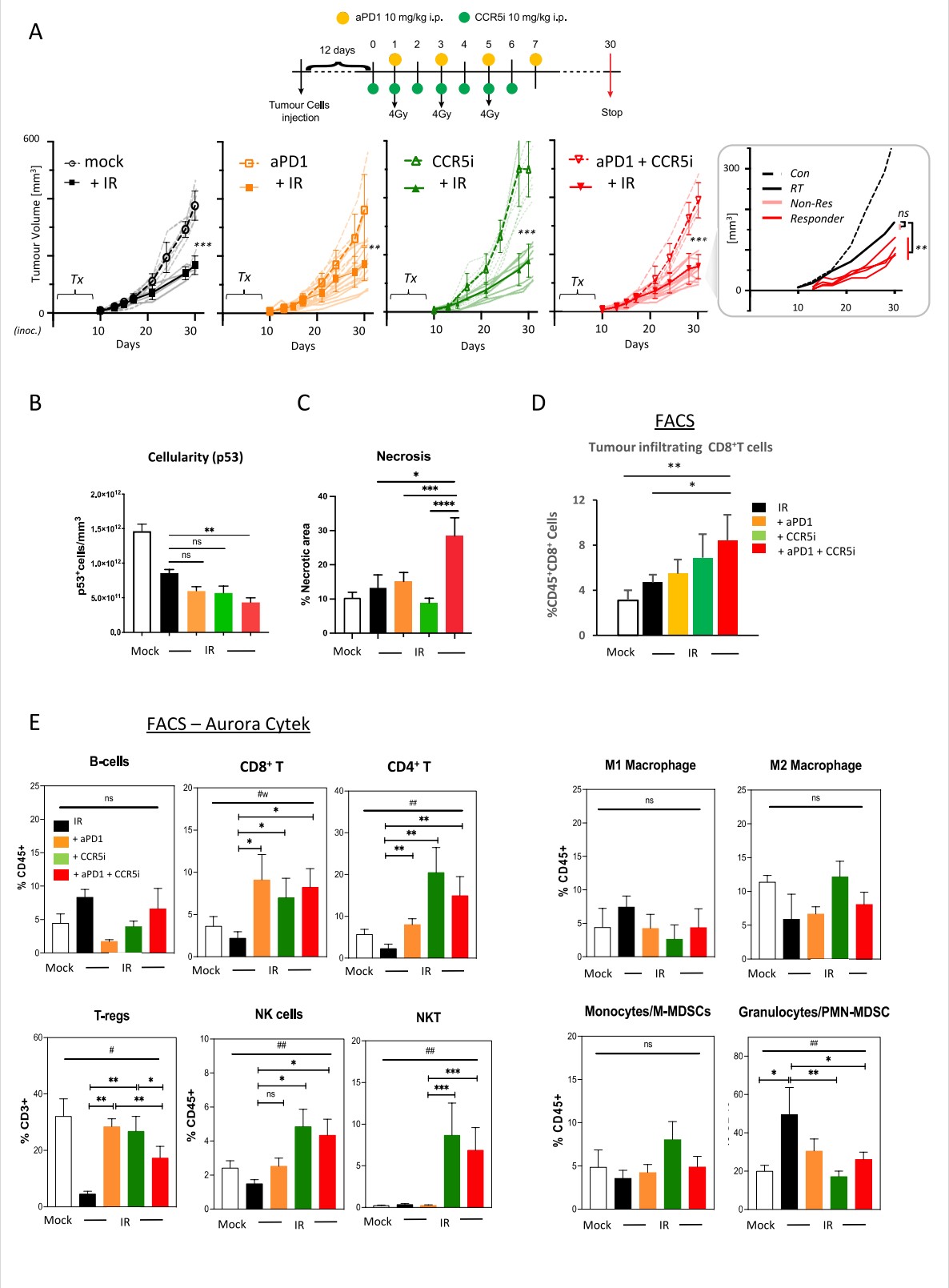

**Figure 3.** Immune profiling of pancreatic ductal adenocarcinoma (PDAC) orthotopic tumours in response to therapy. (**A**) Timeline of triple treatment regimen (maraviroc, αPD1, and radiotherapy) following orthotopic injection of KPC-F cells. A total of 12 days post-orthotopic injection of 500 KPC-F cells in the pancreas of wildtype C57BL/6 mice, mice were treated as follows: 7 consecutive days of 10 mg/kg intraperitoneal injection of maraviroc and 4 alternating days of 10 mg/kg intraperitoneal injection of αPD1. Mice were followed for up to 30 days following the start of the treatment regimen.

*Figure 3 continued on next page*

*Figure 3 continued*

Tumor volumes were measured by MRI and growth curves of individual treatment groups are plotted with or without radiotherapy as measured by MRI. Average growth curves ± SD are depicted in bold, individual mice are shaded (without IR; dashed, with IR; solid). Insert: expanded view of triple combination to show 'responders' display a significant benefit over RT alone. (**B**) Quantification of pancreatic tumors derived from (**A**) stained by IHC for p53. (**C**) Quantification of necrotic areas in pancreatic tumors derived from (**A**) based on H&E staining. (**D**) Quantification of infiltrating CD8 T-cells in pancreatic tumors derived from (**A**) by flow cytometry. (**E**) Profiling of infiltrating immune cells in pancreatic tumors derived from (**A**) by flow cytometry as in *Figure 2B*. Single, live cells were included for analysis and are represented as frequencies of Live/CD45$^+$ cells or total CD3$^+$ for FoxP3$^+$ Tregs. Significance was tested using the Welch and Brown-Forsythe ANOVA for parametric data. #p<0.05, ##p<0.01, or Kruskal-Wallis test for non-parametric data #p<0.01; pairwise comparisons (Student's t-test). *p<0.05, **p<0.01, ***p<0.005, ****p<0.001.

The online version of this article includes the following figure supplement(s) for figure 3:

**Figure supplement 1.** Accessory data to support pancreatic ductal adenocarcinoma (PDAC) tumour model immune monitoring.

*supplement 1A*). As observed with flow cytometric and immunohistochemical analyses, multiplex staining of tumor sections also revealed a significant increase of CD8$^+$ T-cells per mm$^3$ when IR was used in combination with CCR5i or CCR5i+αPD1 (*Figure 4—figure supplement 1B*). Notably, despite having responders and non-responders in the combination groups (*Figure 4A*), a significant overall reduction in E-cadherin$^+$ tumor cells as a percentage of total DAPI$^+$ cells was observed (*Figure 4—figure supplement 1C*), supporting a decrease in cellularity and an increase in tumor necrosis with the IR/CCR5i/αPD1 combination (*Figure 3B and C*). The increase in CD8$^+$ T-cells with combination treatment appears independent of tumor area and is matched by a similar increase in NK cells (*Figure 4—figure supplement 1B and C*). To correlate immune infiltration against loss of tumor cells (a measure of local tumor control), we determined relationships between CD4 T-, CD8 T-, and NK cell populations and E-cadherin$^+$ cells across all tumor sections (independent of treatment) and found a significant inverse correlation for both CD8 T- and NK cells (r$^2$=−0.3, p=0.038 and r$^2$=−0.33, p=0.026, respectively), but not CD4 T-cells (*Figure 4B*).

We next wondered whether the association between NK cells and tumor control was due to cytotoxicity of functionally active NK cells by focusing our analysis on the NKG2D$^+$ NK cell population, given that this is one of the major activating receptors on NK cells and is required for the lysis of target cells (*Bryceson and Ljunggren, 2008*). Surprisingly, the proportion of infiltrating CD3$^-$NK1.1$^+$NKG2D$^+$ cells (NK$^{Active}$) in stained tissue sections was reduced under IR/CCR5iαPD1 combination treatment, implying a decrease in total NK cell cytotoxic capacity (*Figure 4—figure supplement 1D*). The reduced NKG2D expression on NK cells may be a result of the prolonged engagement by ligands expressed on tumor cells, followed by ligand-induced endocytosis and degradation (*Quatrini et al., 2015*), or the shedding of NKG2D ligands by tumor cells (*Kaiser et al., 2007*). In both instances, receptor downregulation causes a reduction in cytotoxicity and impairs NK cell responsiveness to tumor cells, potentially contributing to exhaustion (*Groh et al., 2002*). Moreover, circulating NK cells from PDAC patients show reduced NKG2D levels compared to healthy controls (*Peng et al., 2013*) supporting the notion that chronic exposure to NKG2D ligands expressed or shed by PDAC cells might cause NKG2D down-modulation and a hypo-responsive phenotype. Indeed, the reciprocal CD3$^-$NK1.1$^+$NKG2D$^-$ population increases upon triple combination treatment (*Figure 4—figure supplement 1D*). Surprisingly, the correlation of NK cells with decreased frequency of E-cadherin$^+$ cells was completely abrogated with selection for more functionally competent NK cells (NK$^{Active}$), implying that the opposite hypoactive NKG2D$^-$ NK population (NK$^{NKG2D-ve}$) is responsible for correlation of NK cells with E-cadherin loss and indeed, a superior inverse correlation is observed for NK$^{NKG2D-ve}$ and E-cadherin compared to NK$^{Active}$ or NK$^{Total}$ (r$^2$=0.52, p=0.003) (*Figure 4C*). To explore this association is relation to tumor control, sections were split into either E-cadherin$^{high}$ or E-cadherin$^{low}$ and the extent of immune involvement was represented as a percentage of total DAPI$^+$. While CD8 T-cells significantly segregated with E-cadherin$^{low}$, implying a contribution to tumor control, the association of NK$^{NKG2D-ve}$ was vastly more significant (*Figure 4D*).

To further dissect the association between NK cells and loss of tumor cells, we immunophenotyped NK cells infiltrating tumors of mice treated with the IR/CCR5i/αPD1 regimen with an NK cell dedicated spectral flow cytometry panel. Using a combination of uniform manifold approximation and projection (UMAP) dimensionality reduction and manual gating, we found that more than two-thirds of tumor-infiltrating NK cells expressed the canonical markers of tissue residency CD103 and CD49a (*Figure 4—figure supplement 1E and F*), with the double positive population (hereafter referred

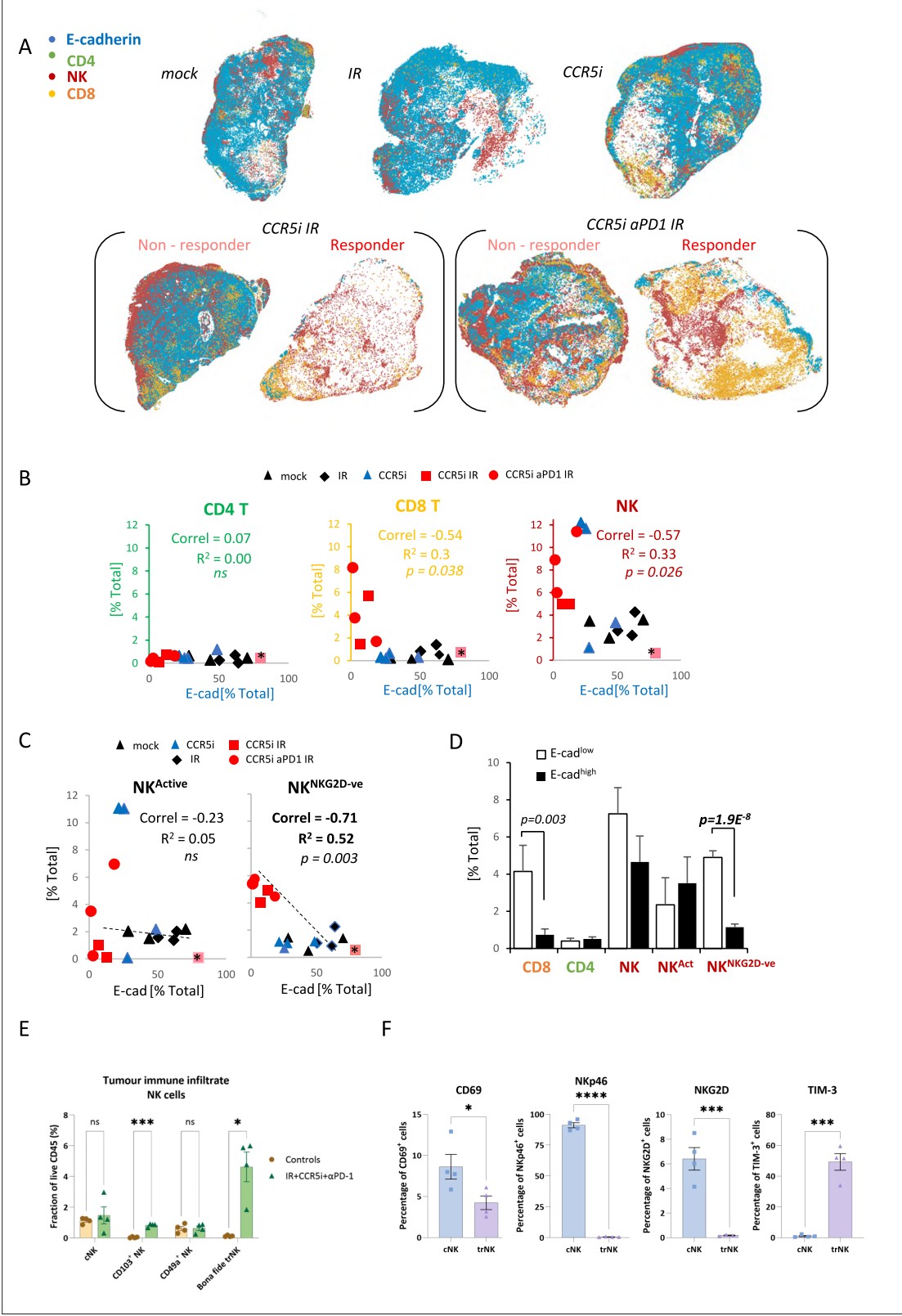

**Figure 4.** Non-cytotoxic natural killer (NK) cells associate with tumor control. (**A**) Spatial plots of individual cells identified using HALO software of scanned multiplex immunofluorescence murine orthotopic pancreatic tumor slices. Positive staining is identified as the marker of interest and DAPI[+] (nucleus stain) signal. Responders and non-responders to treatment are based on loss of E-cadherin staining. (**B**) Correlations of total CD4 T (CD3[+]CD8[-]), CD8 T (CD3[+]CD8[+]), and NK cells (CD3[-]NK1.1[+]) plotted against %positive E-cadherin[+] cells as derived from (**A**). (**C**) Correlation of %positive

*Figure 4 continued on next page*

*Figure 4 continued*

segregated NK cells plotted against %positive E-cadherin$^+$ cells as derived from (**A**). K cells were segregated based on expression of NKG2D; NK$^{Active}$; NK1.1$^+$NKG2D$^+$; NK$^{NKG2D-ve}$; NK1.1$^+$NKG2D$^-$. (**D**) Intra-tumoral immune cells of stratified pancreatic tumors based on low or high E-cadherin percentage (cut-off: 20%). Significance was tested for p<0.05 with a two-tailed Student's t-test. *Censored non-responder. (**E**) Proportion of infiltrating tissue-resident NK (trNK) (Live/CD45$^+$CD3$^-$CD19$^-$NK1.1$^+$CD103$^+$CD49a$^+$), conventional NK cells (Live/CD45$^+$CD3$^-$CD19$^-$NK1.1$^+$CD103$^-$CD49a$^-$), CD103$^+$ NK, and CD49a$^+$ NK cells isolated from orthotopic pancreatic tumors of mice treated with the IR+IT regimen and controls, as a percentage of CD45$^+$ cells. Significance was tested for p<0.05 with a Student's t-test. (**F**) Comparative surface expression of activation marker (CD69), activating receptors (NKp46, NKG2D), and exhaustion marker (TIM-3) on conventional NK (cNK) cells and trNK cells isolated from orthotopic pancreatic tumors. Significance was tested for p<0.05 with a Student's t-test. *p<0.05, ***p<0.005, ****p<0.001.

The online version of this article includes the following figure supplement(s) for figure 4:

**Figure supplement 1.** Identification of a tissue-resident natural killer (trNK) population in mice.

to as bona fide trNK cells) being barely detectable in untreated mice and rising nearly 40-fold after treatment (*Figure 4E*). The remainder cells expressed neither marker of tissue residency and were, therefore, named conventional NK cells.

We then compared these two clusters and found that trNK were less likely to be activated and expressed significantly lower levels of the activating receptors NKG2D and NKp46, as well as significantly higher levels of the inhibitory receptor TIM-3 (*Figure 4F*). In line with our multiplex IF data (*Figure 4—figure supplement 1D*), these data indicate that NK cells infiltrating tumors of mice treated with the IR/CCR5i/αPD1 combination were predominantly tissue resident and hypoactive, with a minority displaying a more conventional, fully active phenotype.

## Heterogenous subsets of NK cells in PDAC exhibit differential inhibitory and activating signatures

To extrapolate our findings to the human setting, we explored scRNA-seq data from isolated NK and T-cells derived from human pancreatic cancer patients. A total of 51,561 cells were catalogued into 17 distinct cell lineages annotated with canonical gene signatures as described in Steele et al. using unsupervised clustering (*Steele et al., 2020*; *Figure 5—figure supplement 1*). UMAP of the lymphocyte compartment did not delineate NK subpopulations and showed overlap with CD8 T-cells (*Figure 5—figure supplement 1*). Therefore, we focused on UMAP of the CD8 and NK compartment alone, which led to three clear NK subpopulations that are distinct from CD8 T-cells and clearly separate out in clusters of hypoactive (NK_C1), fully active (NK_C2), and cytotoxic (NK_C3) NK cells from T-cells (*Figure 5A*). In agreement with downregulation of circulating NKG2D$^+$ in PDAC patients, NKG2D (*KLRK1*) expression was below detection across all CD8 and NK subpopulations retrieved from patient tumors (*Groh et al., 2002*). As expected, CD16$^{high}$ (*FCGR3a*) cytolytic NK cells (NK_C3) cluster separately from CD16$^-$ NK cells (NK_C1) which are more enriched for genes related to cytokine secretion than cytolytic function (*Figure 5D*, *Figure 5—figure supplement 1*). The NK_C1 cluster correlates best with the tissue-resident, hypoactive NK phenotype observed in mice as they similarly displayed reduced cytolytic (reduced *NKG7*, *NKp80*, *GZMA*, and *PRF1*) with additional expression of tissue residency markers *CD103*, *CD49a*, and the adaptive activating receptor NKG2C (*KLRC2*) (*Figure 5B and C*). While adaptive NK cells in the peripheral circulation are associated with a CD56$^{dim}$CD16$^+$ phenotype, adaptive-like (i.e. NKG2C$^+$) CD56$^{bright}$CD16$^{dim}$ trNK cells have been identified in other epithelial sites such as the lungs (*Brownlie et al., 2021*), therefore, our pancreatic tissue NK_C1 cells could comprise, at least in part, adaptive-like trNK cells. Downstream UMAP analysis of the NK cell population distinguishes these three subsets of NK cells (NK_C1, NK_C2, and NK_C3) based on 41 differentially expressed genes (*Figure 5D and E*).

Given that CD56 expression correlates with increased survival in PDAC patients (*Figure 1C*), we were intrigued to notice that the hypoactive NK_C1 cluster is enriched for CD56 but not CD16 expression (*Figure 5D*). These data suggest that the NK_C1 cluster represents a subset of NK cells in PDAC that display tissue residency markers (*ITGAE*, CD103 and *ITGA1*, CD49a) and have an immune-secretory phenotype (XCL1) as opposed to cytotoxic phenotype (NK_C2 CD16$^+$, NKG7$^+$) (*Figure 5E*). Furthermore, we defined a core signature gene set to distinguish NK_C1 from other NK subpopulations (*Figure 5F*). Verification of the NK_C1 population as trNK cells in a second scRNA-seq pancreatic cancer dataset of *Peng et al., 2013*, further supports the existence of these hypoactive cells in an independent PDAC cohort (*Figure 5—figure supplement 1E and F*).

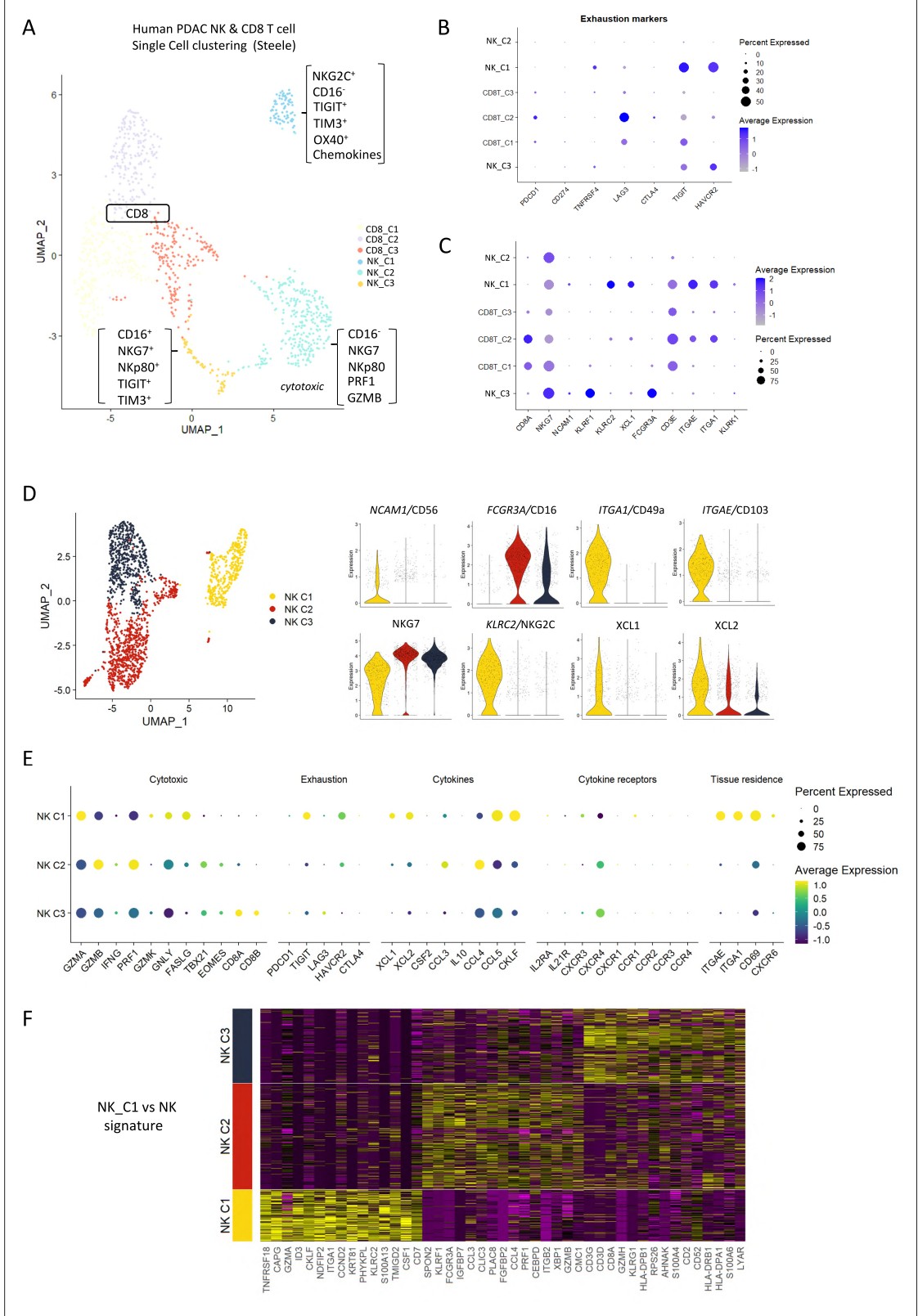

**Figure 5.** Identification of a tissue-resident natural killer (trNK) signature from single cell data of pancreatic ductal adenocarcinoma (PDAC) patients. (**A**) Uniform manifold approximation and projection (UMAP) of the CD8⁺ T and natural killer (NK) sub-clusters from Steele et al. (**B**) Dot plot showing the expression of exhaustion-related genes across CD8⁺ T- and NK cell sub-clusters. (**C**) Dot plot showing highly expressed genes for each sub-cluster. (**D**) UMAP of the three NK sub-clusters (left), and violin plots comparing the expression of NK subtype-associated genes between the sub-clusters (right).

*Figure 5 continued on next page*

*Figure 5 continued*

(**E**) Dot plot showing the different gene expression programs across the three NK sub-clusters. (**F**) Heatmap showing the top 15 upregulated markers for each NK sub-cluster compared against total NK cluster.

The online version of this article includes the following figure supplement(s) for figure 5:

**Figure supplement 1.** Human single cell data indicating a non-cytotoxic population of natural killer (NK) cells.

## trNK cells in PDAC show differential communication

The NK_C1 population had the highest expression of the chemokines XCL1 and XCL2, which have been demonstrated to attract cDC1 (**Böttcher et al., 2018**) and thereby increase cross-presentation to CD8+ T-cells. We next explored myeloid populations in the Steele et al. scPDAC dataset (**Figure 6—figure supplement 1A**) and identified cDC1 cells as XCR1+ enriched compared to other DC subsets (**Figure 6A**, **Figure 6—figure supplement 1B**). As XCR1 is the receptor for XCL1/2, this suggests that one of the main directions of communication from NK_C1 cells in PDAC is to cDC1 (**Figure 6B**). We next employed the R package 'CellChat' (**Jin et al., 2021**) to further dissect the crosstalk between NK_C1 and other cells found in the pancreatic tumor microenvironment. We again observed that NK_C1-derived XCL1/2 is the strongest signal to XCR1+ cDC1s (**Figure 6C**), supporting active cross-talk between NK_C1 and cDC1. Analysis of specific ligand-receptor pairs between NK_C1 and 24 other cell groups yielded 32 significant interactions, of which the *TNFSF14* (LIGHT)-*TNFRSF14* pair was the most universal intercellular interaction, whereas CD74 (or *MIF*) signaling showed the highest communication probability (**Figure 6—figure supplement 1C**).

Within the immune cell compartment, communication signals between NK_C1 and macrophages were the most abundant and diverse, followed by cDCs (cDC2 and cDC1) and Tregs (**Figure 6—figure supplement 1C**). Similarly, cDC1 in these PDAC samples also seem capable of presenting MHC class I peptides to CD8+ T-cells, but surprisingly also MHC class II to CD4+ Tregs (**Figure 6—figure supplement 1D**). Next, significant receptor-ligand interactions were segregated as 'outgoing' or 'incoming' signals to understand directionality of communication. Among the outgoing signals from different cell types, NK_C1 cells contributed the highest IL-16, CSF, LIGHT (TNFSF14), FASLG, and MIF signals in tumors (**Figure 6D**). IL-16 is mainly known as a chemoattractant for CD4+ T-cells, however CD4+ dendritic cells have also been demonstrated to be recruited by IL-16 (**Bialecki et al., 2011**; **Kaser et al., 1999**; **Vremec et al., 2000**). IL-16 has also been described to increase HLA-DR levels in CD4+ T-cells and eosinophils, indicating that it has the potential to induce MHC antigen presentation in CD4+ cells (**Cruikshank et al., 1987**; **Rand et al., 1991**). Taken together, these interactions support the hypothesis that trNK cells may improve tumor control via recruitment of cDC1 via XCR1, while promoting DC maturation via LIGHT-CD86 signaling (**Zou and Hu, 2005**) and supporting DC antigen presentation to both CD4+ and CD8+ T-cells via MIF-CD74 signaling (**Basha et al., 2012**; **Figure 6E**). On the other hand, dendritic cell-secreted BAG6 could promote both survival and cytokine release of NK cells by binding to NKp30 (**Simhadri et al., 2008**) and directly signal to CD8 T-cells via CXCL16-CXCR6 (**Di Pilato et al., 2021**; **Vella et al., 2021**), thereby generating an anti-tumor feedforward loop (**Figure 6E**). IL-16 also communicates to CD4+ Tregs, which could contribute to immunosuppression via enhanced Treg migration and expansion (**Figure 6D**; **McFadden et al., 2007**). However, Tregs may also be susceptible to Fas-mediated cell death due to NK_C1 expression of Fas ligand (**Figure 6D**). This could enhance immune cell infiltration and revoke the immunosuppressive environment, ultimately contributing to increased tumor control.

These results so far suggest the presence of immunoregulatory trNK cells in PDAC that are involved in an intricate immune communication network with DCs and CD8 T-cells to enhance anti-tumor immunity. To support this hypothesis, we explored the correlation between trNK and CD8 T infiltration in our murine orthotopic tumor model and found a highly significant positive association ($R^2=0.6571$, p=0.003), which was strengthened when we focused on untreated (Mock) vs IR+IT combinations where trNK cells are evident ($R^2=0.9223$, p=0.0004) (**Figure 6F**). To ascertain if this also holds true in human PDAC, we first specified the NK_C1 signature as a 14-gene signature that was specific for our trNK cells over all other cells to interrogate these cells in bulk datasets (**Figure 6—source data 1**). We next explored the CD8 T:trNK cell relationship in PAAD_TCGA and, as the model in **Figure 6E** predicts, this interaction to be cDC1 dependent, binned the PAAD_TCGA cohort into quartiles based on differential cDC1 signature expression to test dependence of CD8 T:trNK on cDC1

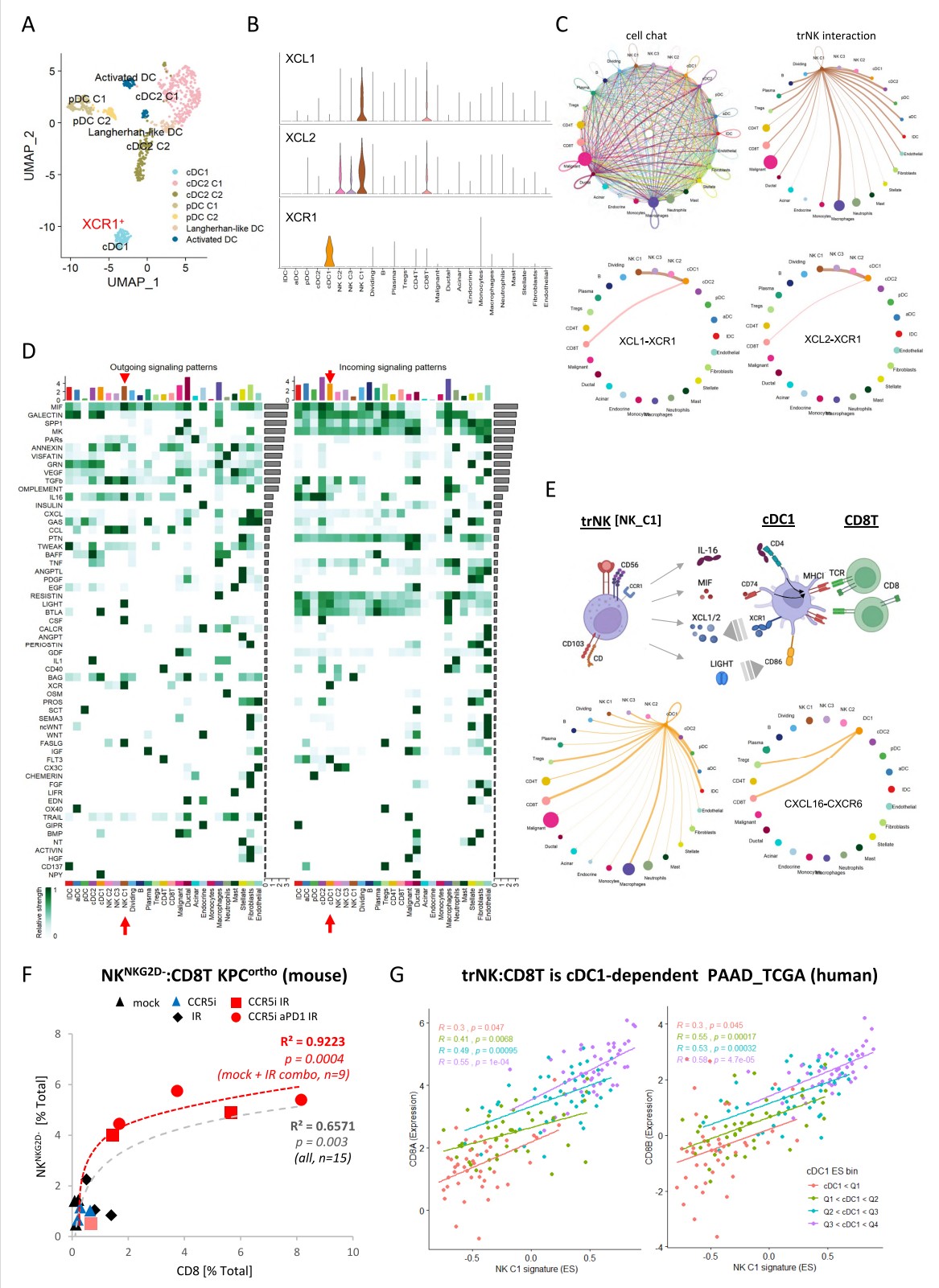

**Figure 6.** CellChat communication indicates tissue-resident natural killer (trNK) communicate directly to cDCs. (**A**) Uniform manifold approximation and projection (UMAP) of the dendritic cell sub-clusters from the Steele dataset. (**B**) Violin plot showing the expression of XCL1, XCL2, and XCR1 across all cell types. (**C**) Circle plots showing interactions across all cell types (top left), signals coming from the tissue-resident natural killer (trNK) cells (top right), the XCL1-XCR1 interaction (bottom left), and the XCL2-XCR1 interaction (bottom right). The width of edges represents the communication strength.

*Figure 6 continued on next page*

Figure 6 continued

(**D**) Heatmap showing the summary of secreted signaling communications of outgoing (left) and incoming (right) signals. The color bar represents the communication strength, and the size of the bars represents the sum of signaling strength for each pathway or cell type. (**E**) Schematic overview of the trNK to type 1 conventional dendritic cell (cDC1) and cDC1 to CD8 T-cell communication axis (top). Circle plots of all outgoing signals from cDC1 (bottom left) and the CXCL16-CXCR6 signaling (bottom right). (**F**) Correlation of HALO data on total NK$^{NKG2D-ve}$ NK cells (CD3$^-$NK1.1$^+$NKG2D$^-$) with CD8 T-cells (CD3$^+$CD8$^+$) from stained sections of treated KPC_F orthotopic tumors, R$^2$ and p-values indicate positive correlation across all tumors (gray, n=15) or limited to mock, CCR5i+IR and IR/CCR5i/αPD1 combination (red, n=9). (**G**) Correlation of trNK signature with CD8A (left) or CD8B (right) in bulk RNA-seq from TCGA_PAAD and binned into quartiles based on extent of cDC1 involvement as assumed by cDC1 signature (*Figure 6—source data 1*).

The online version of this article includes the following source data and figure supplement(s) for figure 6:

**Source data 1.** Gene signatures for trNK and cDC1.

**Figure supplement 1.** Profiling of dendritic cells indicates specific communication.

levels (*Figure 6G*). PDAC patient tumors with the lowest evidence for cDC1 involvement had a weak correlation of trNK_C1 with CD8A and CD8B (p=0.047; p=0.045, respectively) which rises to a strong highly significant correlation when the highest levels of cDC1s are present (R=0.55, trNK_C1 vs CD8A p=0.0001; R=0.58, trNK_C1 vs CD8B p=0.000047) (*Figure 6G*).

## NK cell signature correlates with improved survival

As the presence of trNK and correlation with CD8 T-cells appears boosted by IR, we next further explored our original finding that CD56 correlates with highly significant survival in PDAC (*Figure 1*, *Figure 1—figure supplement 1*). We hypothesized that tumors with a high *NCAM1*/CD56 signature may reflect a greater indication of NK cell recruitment, including a high proportion of trNK (CD56$^{bright}$CD16$^{low}$), whereas tumors with a low CD56 signature might have a lower proportion of infiltrating (tr) NK cells and, therefore, could benefit from IR. Indeed, separating PAAD_TCGA patients into those who received radiotherapy (RTx) vs those that did not showed that the benefit in overall survival of PDAC patients is only apparent in the CD56$^{low}$ patient group (log-rank p<0.0001, *Figure 7*). We next explored whether CD56-associated survival is specifically due to the presence of trNK cells using our NK_C1 signature (*Figure 6—source data 1*). Analysis of primary PDAC tumors from the TCGA (TCGA_PAAD) demonstrates that patients with tumors enriched for the trNK cell (NK_C1) gene signature were associated with improved PDAC survival compared to patients without trNK involvement (*Figure 7*, *Figure 7—figure supplement 1A*). Similarly to CD56$^{low}$ patients, we find that patients with trNK$^{low}$ benefit from RTx (log-rank p<0.005, *Figure 7*). Notably, we find that despite an overall poor prognosis, CCL5$^{high}$ patients (TCGA and CPTAC3) are significantly enriched for the trNK_C1 gene signature (*Figure 7*), potentially due to strong CCR5 or CCR1-mediated recruitment (*Figure 1—figure supplement 1A*). However, as CCL5 also recruits MDSCs, TAMs, Tregs, and conventional NK cells via CCR5, CCR5i could prevent a tumor suppressive microenvironment, both directly and indirectly by retaining trNK cells that remove Tregs through FASL (*Figure 7—figure supplement 1*). Therefore, we would expect that patients with CCL5$^{high}$ NK_C1$^{low}$ would perform significantly worse than patients with CCL5$^{high}$NK_C1$^{high}$. Indeed, CCL5$^{high}$ patients enriched for the NK_C1 signature or CD56 had significantly improved overall and disease-free survival (*Figure 7*). This also supports a model where CCL5-mediated recruitment of NK cells can be beneficial in the absence of CCL5-CCR5 recruitment, likely via CCR1 signaling (*Ajuebor et al., 2007*; *Figure 7—figure supplement 1*, *Figure 1—figure supplement 1A*). These results suggest that despite high CCL5 levels and an overall poor prognostic outcome, the presence of trNK cells significantly improves survival and provides an opportunity for intervention with the IR+IT combination.

Finally, we explored whether the NK_C1 gene signature could be a prognostic marker in solid malignancies other than PDAC by expanding our TCGA analysis. The majority of cancers with enriched expression of our NK_C1 gene signature showed improved survival (*Figure 8*) (apart from endometrium and prostate) as a continuous variable or as a high vs low in a univariate Cox regression (*Figure 8—figure supplement 1*). The latter result supports our hypothesis that enrichment of trNK cells is a protective factor across most solid malignancies. Moreover, the ability to enrich for this population with IR+IT combinatorial strategies supports the idea that trNK cells generally improve overall survival though improving CD8 T-cell activity in solid cancers.

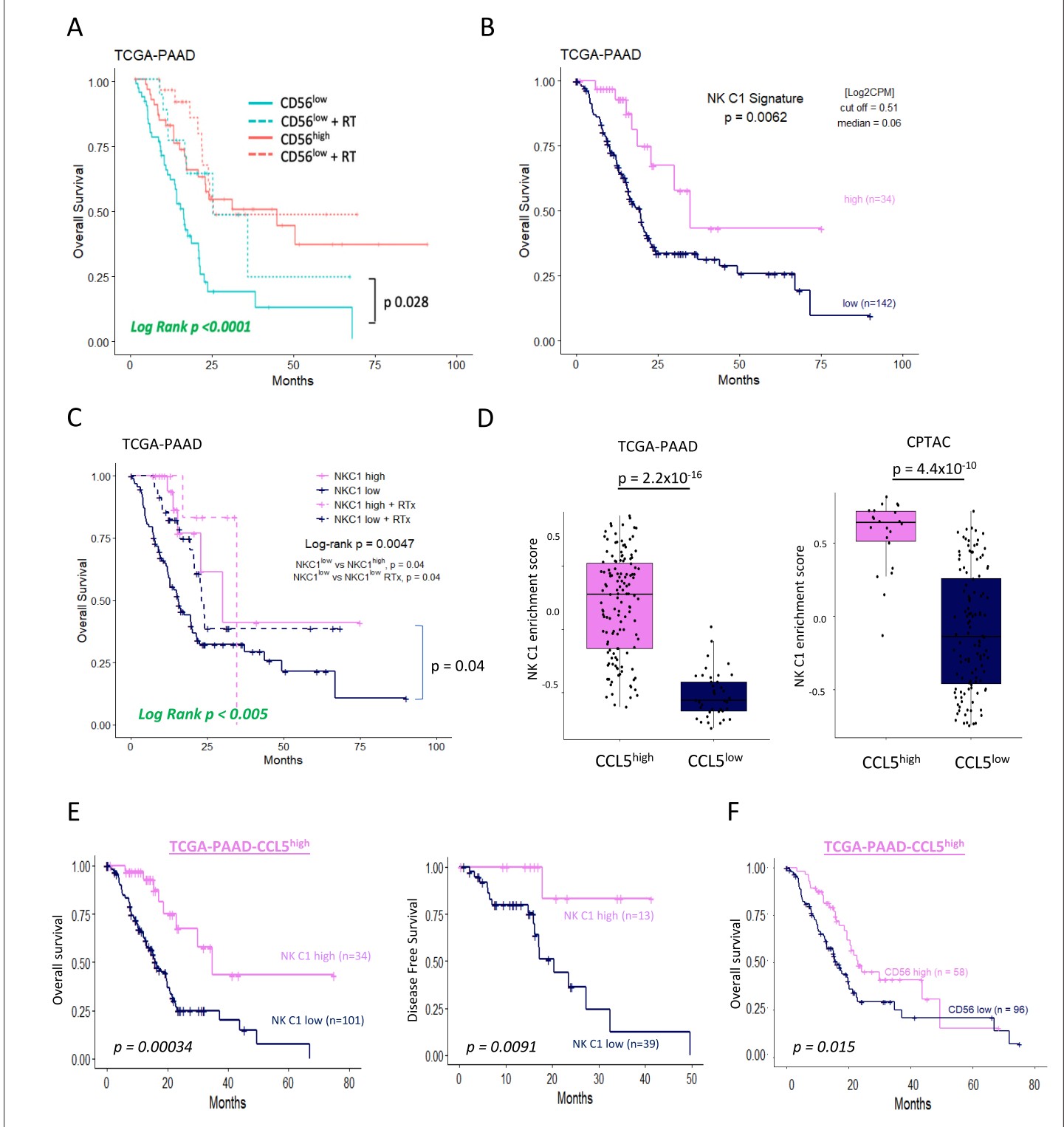

**Figure 7.** Tissue-resident natural killer (trNK) associate with better survival in pancreatic ductal adenocarcinoma (PDAC).

The online version of this article includes the following figure supplement(s) for figure 7:

**Figure supplement 1.** Model for CCL5 mediated recruitment of immune cells in the presence of CCR5i.

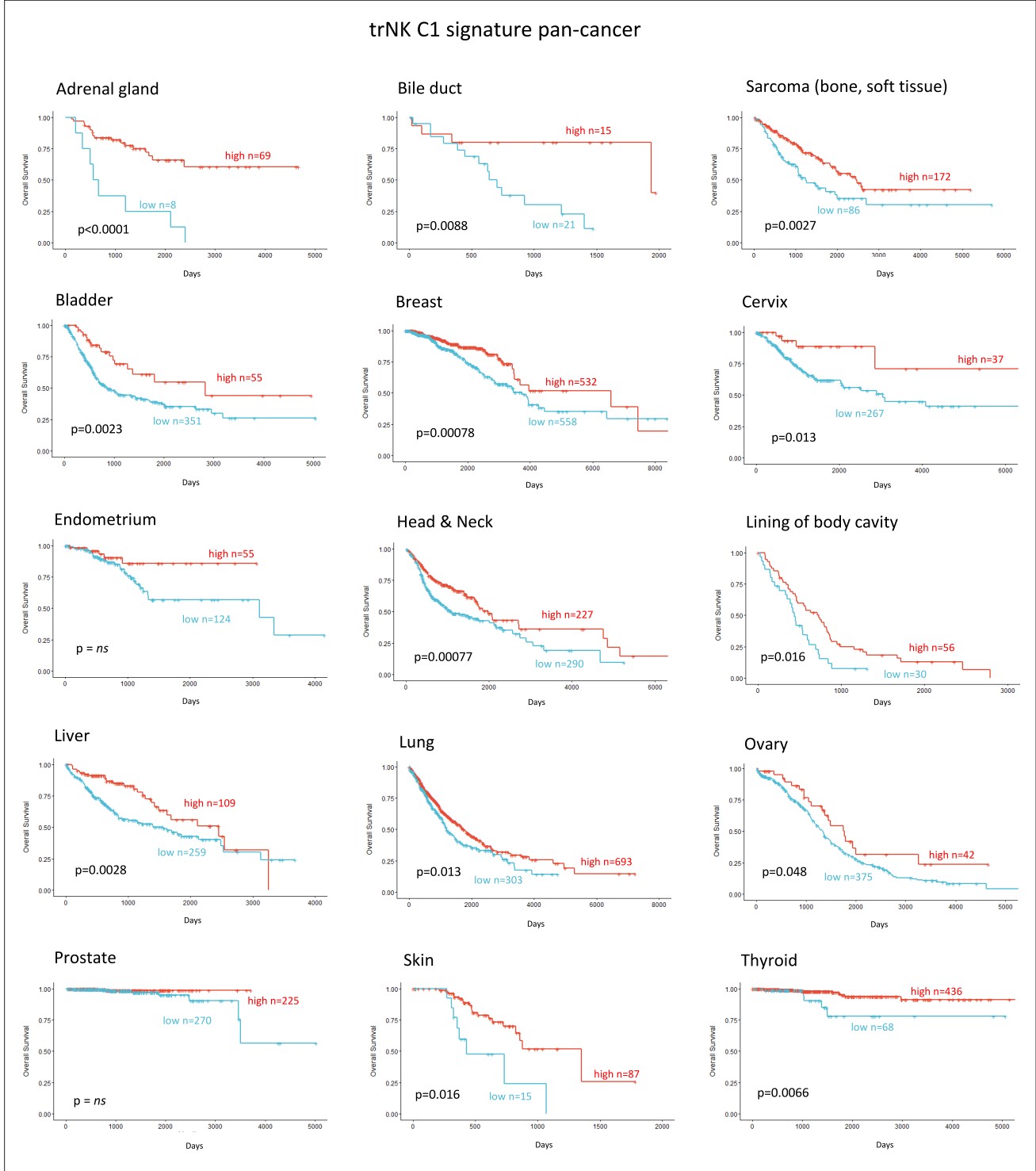

**Figure 8.** Tissue-resident natural killer signature associates with survival pan cancer.

The online version of this article includes the following figure supplement(s) for figure 8:

**Figure supplement 1.** Tissue-resident natural killer cell signature survival hazard ratios.

# Discussion

Novel approaches such as immunotherapies struggle to improve outcome in PDAC as tumors are stromal-rich, myeloid-involved immunosuppressive microenvironments devoid of cytotoxic lymphocytes (*Chouari et al., 2023*). Most strategies to combat immune suppression, e.g., targeting inhibitory checkpoints such as PD1/PDL1, fail as single therapies because PDAC is largely devoid of the CD8 T-cells as these agents are designed to reactivate. Combining them with inhibitors of myeloid suppression (e.g. CXCR2) to increase CD8 T penetration also failed to impact survival in clinical trials despite showing promise in preclinical studies (*Steele et al., 2015*, *Siolas et al., 2020*). Similarly, high-dose hypo-fractionated ablative radiotherapy has been employed to create an acute localized inflammatory response to stimulate intra-tumoral penetration of CD8 T-cells, but even in conjunction with PD1/PDL1 blockade this approach has not shown benefit in PDAC (*Parikh et al., 2021*). Novel strategies in preclinical PDAC models comprising IR, αPD1, and BMS-687681 (a dual CCR2/CCR5i) have shown promising increases in CD8 T-cells (*Wang et al., 2022*), whereas the combination of IR and the bifunctional agent αPD-1/IL-2Rβγ stimulates tumor penetration of polyfunctional stem-like activated CD8 T-cells and DNAM1$^+$ cytotoxic NK cells (*Piper et al., 2023*). Together, these approaches indicate potential benefits of a coordinated alteration of the suppressive microenvironment and checkpoint blockade to reduce Tregs and increase NK cell infiltration in addition to supporting CD8 T-cell activity. The utility of RT for localized PDAC has been controversial, but innovative technology can now deliver IR at higher doses with greater precision (*Mills et al., 2022*). Rather than simple ablative radiotherapy, this strategy is being employed to increase localized damage that stimulates an acute damage response in immune-cold tumors or increase tumor neoantigens to stimulate adaptive responses (*Sodergren et al., 2020*). In support of this, the use of IR alongside αPD-1/IL-2Rβγ or FAKi appeared to induce durable immunity in preclinical models (*Lander et al., 2022*; *Piper et al., 2023*), suggesting the induction of immunologic memory against tumor antigens is possible.

We were led by our previous clinical trial results implicating serum CCL5 levels as a negative prognostic marker for PDAC survival, which we now validate in two independent validation datasets. CCL5 has both pro-inflammatory roles as a chemoattractant for leukocytes and anti-inflammatory activity via recruitment of CCR5$^+$ Tregs (*Tan et al., 2009*). To maintain beneficial signals but limit pro-tumorigenic signaling, we targeted CCR5 alone using maraviroc. Not surprisingly, in an orthotopic KPC model of PDAC, we found that Tregs were restricted by CCR5i directly and the combination of IR and IT (CCR5i, αPD1, or CCR5i/αPD1) correlates with a progressive enrichment of CD8 T-cells. Intriguingly, CD8 T-cells alone were insufficient to explain the loss of cellularity or the increase in necrotic tumors seen with the IR/CCR5i/αPD1 combination, but we do observe a significant increase in total NK and NKT cells that correlates with better local tumor control.

NK infiltration has been associated with positive outcomes in many solid tumors, considered to be due to the positive impact of cytotoxic NK cells in cytotoxic CD8 T-mediated tumor clearance (*Nersesian et al., 2021*). However, the data supporting the independent contribution of NK activity is difficult to discern due to overlap of expression profiles of cytotoxic cells. Surprisingly, markers of cytotoxic cells or specific CD8/CD4 lymphocyte receptors do not perform as indicators of survival from bulk mRNA-seq datasets (*Figure 5*). This may be due to sensitivity issues but, more likely, the presence of cytotoxic cells alone does not necessarily indicate beneficial responses in patients (*Nersesian et al., 2021*). This is supported by the limited efficacy of IT strategies to tumors with a high mutational burden. NK cells represent a variety of subsets defined by surface markers, found in the periphery (circulating), secondary lymphoid organs (spleen, lymph nodes), and specific tissues (e.g. lung, liver, uterus) where markers of tissue residency increase retention or prevent egress (*Hashemi and Malarkannan, 2020*). In addition to trNK cells, tumors appear to accumulate NK populations that become less cytotoxic through downregulation of activating receptors, (NKG2D, NKp46, NKp44) and increased expression of inhibitory receptors (NKG2A) or repression/exhaustion markers (e.g. TIGIT, TIM3) (*Hashemi and Malarkannan, 2020*; *Marcon et al., 2020*). Importantly, trNK or hypoactive NK cell subsets commonly display a reduction in cytotoxicity but become highly active immunomodulatory players via expression of XCL1 and XCL2 (*de Andrade et al., 2019*). In melanoma, NK-mediated expression of XCL1 is crucial for the migration of XCR1$^+$ cDC1 and therefore the non-cytotoxic NK subsets in tumors may be vital for cDC1-mediated cross-presentation of tumor antigens to CD8 T-cells (*Böttcher et al., 2018*). This may explain why levels of CD8 T-cells in tumors or removal of inhibitory signaling is insufficient to gain tumor control. NK-derived FLT3L correlated with increased

cDC1 infiltration and improved overall survival and enhanced αPD1 treatment responses in melanoma (*Barry et al., 2018*), while intra-tumoral NK cells correlate with increased T-cell and dendritic cell infiltration and improved survival in neuroblastoma (*Melaiu et al., 2020*). Although our NK_C1 cells are not enriched for *FLT3LG* (data not shown), the data collectively demonstrate that therapeutic strategies aimed at increasing the presence of cDC1 or NK cells may work in combination to support treatments that induce CD8 T-cell activity.

Our data suggest that combination therapy to stimulate an acute inflammatory response (IR) together with CCR5i/αPD1 (IT) sufficiently modulates the tumor immune microenvironment to improve tumor control. As frequently observed in solid tumors, reduction of cancer cells correlates with intra-tumoral infiltration of NK cells (CD56$^{bright}$CD16$^-$) and improved CD8 T-cell penetration (*Wu et al., 2020*). Surprisingly, our tumors were penetrated by tissue-resident, hypoactive NKG2D$^-$ NK cells, suggesting a population with reduced cytotoxicity (*Figure 4F*, *Figure 4—figure supplement 1*) which we correlated to a similar population (NK_C1) identified from scRNA-seq of human PDAC samples (*Figure 5—figure supplement 1*). This population was CD56$^+$CD16$^-$, in keeping with the apparent beneficial association of CD56 expression with PDAC survival we identified above. NKG2C is a marker of adaptive NK cells during viral infections where they offer a potential memory capability to innate immunity (*Brownlie et al., 2021*; *López-Botet et al., 2023*). Adaptive NK cells were first identified as a subpopulation of blood-derived CD16$^+$ NK cells in response to viral infection but more recently found to be independent of CD16 expression in lung tissue (*Brownlie et al., 2021*). Although we did not specifically examine markers for adaptive NK cells in our mouse model, the expression of NKG2C in the NK_C1 human cluster led us to believe that at least some of these NK cells might be adaptive-like, trNK cells similar to those identified in the lung (*Brownlie et al., 2021*). Notably, these cells were marked with receptors for tissue residency CD103 (*ITGAE*), CD49a (*ITGA1*), chemokine expression (XCL1/2), and lymphocyte exhaustion markers TIM3 (*HAVCR2*, TIGIT, *TNFRSF4*), potentially suggesting conversion of cytotoxic NK cells to an immunomodulatory phenotype that can persist in tissues (*Brownlie et al., 2021*; *Rückert et al., 2022*). scRNA-seq from PDAC tissue confirm that trNKs could mediate recruitment of cDC1s via XCL1-XCR1, but communicate additional signals, including IL-16, LIGHT (*TNFSF14*), and MIF-CD74, which have the potential to upregulate MHC-I expression, antigen presentation, and co-stimulatory molecules to contribute to cross-presentation (*Basha et al., 2012*; *Cruikshank et al., 1987*; *Zou and Hu, 2005*). We also find that trNK may also recruit CD4$^+$ Tregs via IL-16 but concomitant FAS signaling would lead to Treg apoptosis and support tumor control (*Figure 6D*, *Figure 7—figure supplement 1*).

Strikingly, using our trNK signature we find that elevated involvement of trNKs in PDAC correlates with CD8 T-cell recruitment in a cDC1-dependent manner (*Figure 6G*). Moreover, this supports a model where the inactivation of cytotoxic NK cells in tumors appears to be a conversion to immunomodulatory trNKs and perhaps an important mechanistic switch from innate to adaptive immunity. Our therapeutic combination strategy of short ablative radiation-induced damage (IR) followed by CCR5i/αPD1 (IT) offers a potential regimen to increase disease-free survival in PDAC. Likewise, trNK-like cells with similar traits are increasingly being observed across a variety of cancers (*Brownlie et al., 2023*; *Kirchhammer et al., 2022*; *Marquardt et al., 2015*). Finally, our trNK signature identifies patients with a significant survival benefit across 14 tumor types, indicating a universal phenomenon attributable to this hypoactive, tissue-resident, immunomodulatory NK subset.

# Materials and methods
## Mice and cell culture

Female wildtype C57BL/6 mice (6–8 weeks) were purchased from Charles River (Kent, UK) and maintained in accordance with the UK Animal Law (Scientific Procedures Act 1986). Animal procedures were carried out under local ethics approval. NKPCB-NBC12.1F (KPC-F) cells, derived from a KPC mouse (LSL-Kras$^{G12D}$; LSL-Trp53$^{R172H}$; Pdx1$^{Cre}$) derived from a CD57BL/6 background were cultured in DMEM (Gibco) supplemented with 10% FBS, 2 mM glutamine, 100 U/mL penicillin, 100 µg/mL streptomycin and maintained under 5% CO$_2$, 37°C. The cells were kindly donated by Jennifer Morton (Beatson Institute, Glasgow) and stocks were generated to keep cells at low passage numbers and checked regularly for mycoplasma contamination (Lonza).

## Orthotopic surgery

A total of 500 KPC-F cells (passages 9–11) were injected into female C57BL/6 mice in a volume of 5 µL in a mixture of 94% Matrigel and 6% DMEM (Gibco) using a 26-gauge syringe (Hamilton). Following anesthesia of the mice using 4% isoflurane/oxygen and clipping of hair, an incision of 2 cm was made in the left abdominal site. The spleen and pancreas were exteriorized using forceps and KPC-F cells were injected in the tail of the pancreas. Successful injection was verified by the appearance of a wheal at the injection site with no leakage through the pancreatic capsule. Spleen and pancreas were gently moved back into the peritoneal cavity; the peritoneum was sutured with absorbable 4.0 Vicryl suture (Ethicon) and the skin was closed using a 7 mm wound clip.

## In vivo drug treatments and radiation

Animals were randomized to different treatment groups following orthotopic implementation. For radiation, mice were anesthetized and locally irradiated with 4 Gy using a combination of MRI and the small animal radiation research platform. The CCR5i maraviroc (R&D) was administered intraperitoneally (10 mg/kg in PBS) from day 0 until day 6. Monoclonal Ultra-LEAF purified anti-PD1 antibody (BioLegend) was administered (5 mg/kg in PBS) via IP injection on alternate days from day 1 until day 7. Mice were culled at 30 days or if maximum tumor volume was reached.

## Monitoring of tumor growth via MRI

Mice were anesthetized using 1.5–3% isoflurane, positioned into a custom-made, 3D printed, multi-modality cradle and scanned using a 4.7 T 310 mm horizontal bore Biospec AVANCE III HD preclinical imaging system equipped with 114 nm bore gradient insert (Bruker BioSpin GmbH, Germany). Respiration was maintained at 40–60 breaths/min and monitored using a pneumatic balloon (VX010, Viomedex Ltd, UK) coupled to a pressure transducer and placed against the chest. A threshold-based respiration gating control signal was generated on a custom-built gating device, which allows efficient multi-slice MRI scanning with limited interference of respiratory motion (*Kinchesh et al., 2019*). Tumor volumes were calculated using ITK-SNAP (*Yushkevich and Gerig, 2016*).

**Table 1.** Aurora spectral panel and antibodies list.

| Antigen | Clone | Fluorochrome | Company, catalogue number | Dilution |
|---|---|---|---|---|
| Live/dead | – | Blue | Thermo Fisher, L23105 | 1:1000 |
| CD45 | 30-F11 | AF532 | Thermo Fisher, 58-0451-80 | 1:50 |
| CD19 | 1D3 | SB436 | Thermo Fisher, 62-0193-80 | 1:50 |
| CD3 | 17A2 | eF450 | BioLegend, 48-0032-80 | 1:50 |
| NK1.1 | PK136 | BV650 | BioLegend, 108735 | 1:25 |
| CD8a | 53–6.7 | BV570 | BioLegend, 100739 | 1:25 |
| CD4 | RM4-5 | BV510 | BD Biosciences, 100553 | 1:100 |
| CD25 | 7D4 | AF647 | BD Biosciences, 563598 | 1:100 |
| FOXP3 | FJK-16s | PE | Thermo Fisher, 12-5773-80 | 1:20 |
| CD206 | C068C2 | BV785 | BD Biosciences, 141729 | 1:25 |
| F4/80 | T45-2342 | BV480 | BD Biosciences, 565635 | 1:100 |
| CD80 | REA983 | APC | Miltenyi Biotech, 130-116-46 | 1:100 |
| CD279 (PD-1) | J43 | PE-Cy7 | Thermo Fisher, 25-9985-82 | 1:25 |
| CD195 (CCR5) | REA354 | VB-FITC | Miltenyi Biotech, 30-105-141 | 1:20 |
| CD11b | M1/70 | AF700 | BioLegend, 101222 | 1:100 |
| Ly6C | HK1.4 | PE-Dazzle | BioLegend, 128043 | 1:200 |
| Ly6G | 1A8 | BV421 | BioLegend, 127627 | 1:25 |

## Blood and tissue collection

Following reaching (humane) endpoints, mice were euthanized by approved Schedule 1 methods. Blood (≤1 mL) was collected via cardiac puncture under anesthesia (3–3.5% isoflurance/oxygen) using a 25-gauge syringe (Thermo Fisher Scientific) coated with anticoagulant ACD buffer (0.5 M glucose, 0.5 M trisodium citrate, 0.5 M citric acid; 100 µL). A total of 700 µL of blood was lysed with 12 mL of 1× RBC Lysis buffer and incubated for 10–15 min on a rotor, centrifuged, and washed with buffer (2% FBS, 1 mM EDTA in PBS). Three transverse sections of orthotopic tumors were taken and used for flow cytometry or fixed and stored in 10% neutral buffered formalin. Tumors were disassociated using a mixture of collagenase I, II, and IV (Worthington) and DNAse (Thermo Fisher Scientific) in HBSS (Gibco), and continuously mixed at 850 rpm for 45 min at 37°C, followed by rigors pipetting and additional dissociation for 50 min at 850 rpm and 37°C. The disassociated tissue was strained through a 70 µm cell strainer and neutralized with DMEM+10% FBS prior to incubation with flow antibodies.

## Multicolor spectral flow cytometry

A total of $1\times10^6$ cells for blood and disassociated tissue were surface stained in a 100 µL volume at 4°C in the dark (see *Table 1* for list of antibodies and optimized dilutions). Cells and reference controls were centrifuged at 500×*g* for 2 min at room temperature and fixed and permeabilized using the FOXP3 Fixation/Permeabilization kit (Thermo Fisher Scientific). Samples were intracellularly stained at 4°C in the dark, washed, re-suspended in buffer and stored at 4°C in the dark prior to acquisition. Samples were acquired on a 4-laser Aurora (Cytek) with SpectroFlo software (Cytek, v2). A minimum of 10,000 events for reference control samples and 100,000 of CD45-gated events were recorded. Samples were acquired on a 3-laser (42-channel, V-16, B-16, R-10) or 4-laser (58-channel, UV-16, V-16, B-16, R-10) Cytek Aurora.

## Immunohistochemistry

A total of 24 hr post-fixation at room temperature, tissues were transferred to 70% ethanol and stored overnight at 4°C. Fixed tissues were overnight processed using the STP120 Spin Tissue Processor (Thermo Fisher Scientific) and embedded the next day in paraffin wax. Tissues were cut into 4 µm sections using a Leica RM215 microtome and adhered onto SuperFrost Ultra Plus Adhesion slides (Thermo Fisher Scientific) before being dried overnight in a 37°C incubator.

## Multiplex immunofluorescence

Multiplex immunofluorescence staining was carried out on 4-µm-thick FFPE sections by the Translation Histopathology Laboratory (THL, University of Oxford). In brief, sections were stained using the OPAL protocol (AKYOA Biosciences) on a Leica BOND RXm Auto-Stainer (Leica, Microsystems). Six consecutive staining cycles were performed using primary antibody-Opal fluorophore pairings detailed in *Table 2*. Primary antibodies were incubated for 30 min and detected using the BOND Polymer (Lecia Biosystems) as per the manufacturer's instructions. In brief, sections were baked, dewaxed with BOND dewax solution, rehydrated with alcohol, and incubated with Epitope Retrieval Solution 1 or 2 (ER1, ER2) (Lecia Biosystems) at 100°C for 20 min. Sections were washed X3 with BOND wash, blocked with peroxidase block (3–4% [vol/vol] hydrogen peroxide) for 5 min and subsequently washed 3× with BOND wash. Primary antibodies were incubated for 30 min, washed as before, and incubated with Anti-Rabbit Poly-HRP IgG for 8 min. Sections were washed twice in BOND wash, once in deionized water prior to Opal antibody incubation for 10 min. Sections were washed three times in deionized

**Table 2.** Multiplex IF panel.

| Antigen | Coupled to | Company, catalogue number | Dilution |
|---|---|---|---|
| NKG2D | Opal 520 | Abcam, ab203353 | 1:600 |
| CD161/NK1.1 | Opal 620 | Abcam, ab234107 | 1:20,000 |
| CD3 | Opal 480 | Abcam, ab5690 | 1:300 |
| CD8 | Opal 570 | Cell Signaling, 98941 | 1:800 |
| E-cadherin | Opal 780 | Cell Signaling, 3195 | 1:500 |

water and finally incubated with spectral DAPI (Akoya Biosciences) and slides mounted with VECTA-SHIELD Vibrance Antifade Mounting Medium (Vector Laboratories). Whole slide multispectral images were obtained on the AKOYA Bioscience Vectra Polaris (scanned at ×20 magnification). Batch analysis and spectral unmixing of the tissues was performed with inForm 2.4.11 software. Batch analyzed multispectral images were fused in HALO AI to produce a spectrally unmixed reconstructed whole tissue image.

## Data analysis and statistics

Flow cytometry data was analyzed using FlowJo software. HALO AI software was used for analysis of multiplex immunofluorescence. Due to variations in staining, thresholds for individual colors were manually determined for individual tissue sections and analysis was carried out blinded fashion.

## Bulk RNA-seq analysis

Gene expression and clinical data were obtained via the TCGAbiolinks (2.24.3) R package for TCGA-PAAD ('Integrated Genomic Characterization of Pancreatic Ductal Adenocarcinoma', 2017), and GDC data portal (https://portal.gdc.cancer.gov/analysis_page?app=Downloads) for CPTAC-3 (*Cao et al., 2021*). Raw read counts were transformed to counts-per-million (CPM) and log-CPM using the 'edgeR' Bioconductor package (3.38.0). Inter-sample variation was normalized using the trimmed mean of M-values (TMM) method of 'edgeR'. The 'GSVA' package (1.44.0) was used to generate the trNK signature (*Figure 6—source data 1*) score for individual patients. Gene signature scores and gene expressions were transformed to categorical groups via the maximally selected rank statistics (maxstat) method of the 'survival' package. Kaplan-Meier plots were performed using the 'survival' (3.3-1), and 'survminer' (0.4.9) R packages and the log-rank test was used to statistically compare survival estimates between groups.

## TCGA pan-cancer analysis

For the pan-cancer TCGA survival analysis, gene expression and clinical data were obtained via the UCSC Xena platform. The trNK signature was analyzed for the individual cancer types for Kaplan-Meier survival similar to above. Univariate COX regression of the NK signature for both overall survival and disease-free survival was performed using the 'ezcox' (1.0.2) package.

## Analysis of human scRNA-seq data

Raw scRNA-seq data from the Steele dataset (*Steele et al., 2020*) were obtained from the NIH GEO database by using the accession number GSE155698. The samples included 16 treatment naïve primary tumors (6 surgical resections and 10 fine-needle biopsy specimens) and 3 non-malignant pancreas samples. All analysis was performed using the 'Seurat' (4.3.0) package. Cells with low number of detected genes (<200), high number of mitochondrial genes (>25%), and high number of gene counts (>100,000) were filtered out from further analysis. Data were integrated to remove batch effects using a reciprocal principal component analysis (rPCA) based on the developer's pipeline (https://satijalab.org/seurat/articles/integration_rpca.html). Following integration, data were scaled, and visualized using UMAP on the top 40 principal components. Cell clusters were identified using the 'FindNeighbors' and 'FindClusters' Seurat functions and annotated based on canonical cell markers. Gene expression differences across cell clusters were visualized using the 'DotPlot' and 'DoHeatmap' Seurat functions. Curated gene sets were calculated using the 'AddModuleScore' and compared using violin plots with the 'VlnPlot' function.

## Inference of cell-cell interactions

The 'CellChat' (1.6.1) R package was used to identify and visualize significant cell-cell interactions based on curated ligand-receptor pairs for both secreted signaling and cell-cell contact categories (*Jin et al., 2021*). Network centrality scores were calculated with the 'netAnalysis_computeCentrality' function and global communication patterns were visualized using the 'netAnalysis_signalingRole_heatmap'. Specific interactions of cells of interest were visualized by circle and dot plots by using the 'netVisual_circle' and 'netVisual_bubble' functions, respectively.

## Packages/tools used for analysis

| Package (version) | Repository | Source or reference |
|---|---|---|
| TCGABiolinks 2.24.3 | Bioconductor | https://bioconductor.org/packages/TCGAbiolinks/; *Colaprico et al., 2008* |
| edgeR 3.38.0 | Bioconductor | https://bioconductor.org/packages/edgeR/; *Chen et al., 2016* |
| GSVA 1.44.0 | Bioconductor | https://bioconductor.org/packages/GSVA/; *Castelo et al., 2004* |
| Survival 3.3-1 | CRAN | https://cran.r-project.org/package=survival; *Therneau et al., 2022* |
| Survminer 0.4.9 | CRAN | https://cran.r-project.org/package=survminer; *Kassambara et al., 2021* |
| Ezcox 1.0.2 | CRAN | https://cran.r-project.org/package=ezcox ; *Wang, 2021* |
| Seurat 4.3.0 | CRAN | https://cran.r-project.org/package=Seurat; *Butler et al., 2022* |
| CellChat 1.6.1 | GitHub | https://github.com/sqjin/CellChat; *Jin, 2023* |
| Tidyverse 2.0.0 | CRAN | https://cran.r-project.org/package=tidyverse; *Wickham, 2023* |

## Acknowledgements

The authors thank the oncology preclinical imaging core (S Smart, V Kersemans, PD Allen, M Hill, and JM Thompson) for providing contracted small animal MRI and radiotherapy services, Biomedical Services (K Watson) and H Xu for technical support. Funding: Kidani Memorial Trust, Pancreatic Cancer UK, Cancer Research UK, Precision Panc (A25233), CRUK Beatson Institute (A31287, A29996), and CRUK Scotland Centre (CTRQQR-2021\100006). This work was supported by Cancer Research UK (CR-UK) grant number CTRQQR-2021\100002, through the Cancer Research UK Oxford Centre.

## Additional information

### Funding

| Funder | Grant reference number | Author |
|---|---|---|
| Kidani Memorial Trust | | Simei Go<br>Simone Lanfredini<br>Eric O Neill |
| NIHR Oxford Biomedical Research Centre | | Helen Ferry |
| Pancreatic Cancer UK | | Simei Go<br>Eric O Neill |
| Cancer Research UK | A25233 | Eric O Neill |
| Beatson Institute for Cancer Research | A31287 | Jennifer Morton |
| Beatson Institute for Cancer Research | A29996 | Jennifer Morton |
| CRUK Scotland Centre | CTRQQR-2021\100006 | Jennifer Morton |
| Cancer Research UK | CTRQQR-2021\100002 | Jennifer Morton |

The funders had no role in study design, data collection and interpretation, or the decision to submit the work for publication.

### Author contributions

Simei Go, Formal analysis, Visualization, Writing – original draft; Constantinos Demetriou, Data curation, Formal analysis, Visualization, Methodology, Writing – original draft; Giampiero Valenzano, Investigation, Methodology, Writing – review and editing; Sophie Hughes, Data curation, Formal analysis, Methodology; Simone Lanfredini, Data curation, Formal analysis, Investigation, Methodology; Helen Ferry, Resources, Software; Edward Arbe-Barnes, Formal analysis; Shivan Sivakumar, Methodology; Rachel Bashford-Rogers, Supervision, Methodology; Mark R Middleton, Funding acquisition; Somnath

Mukherjee, Conceptualization, Project administration; Jennifer Morton, Resources, Supervision, Funding acquisition, Methodology; Keaton Jones, Formal analysis, Supervision, Investigation, Visualization, Methodology, Project administration; Eric O Neill, Conceptualization, Data curation, Formal analysis, Supervision, Funding acquisition, Investigation, Visualization, Methodology, Writing – original draft, Project administration, Writing – review and editing

**Author ORCIDs**
Mark R Middleton ⓘ https://orcid.org/0000-0003-0167-1685
Eric O Neill ⓘ https://orcid.org/0000-0002-0060-6278

**Ethics**
This study was performed in strict accordance with the recommendation in the guide for the care of laboratory animals and the Animals (Scientific Procedures) Act 1986 (UK). All animals were handled according to at University of Oxford Biomedical services and Protocols were approved by Animal Welfare Ethical Review Body under the Personal project licence PF0F1EF6E at appropriate facilities within Oxford the hold establishment licences.

Reviewer #1 (Public Review): https://doi.org/10.7554/eLife.92672.3.sa1
Reviewer #2 (Public Review): https://doi.org/10.7554/eLife.92672.3.sa2
Reviewer #3 (Public Review): https://doi.org/10.7554/eLife.92672.3.sa3
Author response https://doi.org/10.7554/eLife.92672.3.sa4

## Additional files

**Supplementary files**
• MDAR checklist

**Data availability**
The authors confirm that the data supporting the findings of this study are available within the article and as source data. Sources of publicly available data and analysis packages are referenced with the text or methods.

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
