## [Editor Report · eLife assessment]

This **valuable** manuscript provides an interesting account documenting the role of resident CD56(br) NK cells in driving interaction with dendritic cells that attract CD8+ T cells to the pancreas cancer tumor microenvironment (TME). The work **convincingly** illustrates how irradiation combined with CCR5i and PD1 blockade leads to a reduction in pancreatic cancer growth that correlates with a reduction in Treg cells and enhancement of NK and CD8 T cells in the TME. The correlation of NKC1 signature with survival in pancreatic cancer patients is indeed of broader interest regarding potential relevance to other types of cancer.

---

## [Referee Report · Reviewer #1 (Public Review)]

Summary:

The authors demonstrate that the immunosuppressive environment in pancreatic ductal adenocarcinoma (PDAC) can be mitigated by a combination of ionizing radiation (IR), CCR5 inhibition, and PD1 blockade. This combination therapy increases tissue-resident natural killer (trNK) cells that facilitate CD8 T cell activity, resulting in a reduction of E-cadherin positive tumor cells. They identify a specific "hypofunctional" NK cell population in both mouse and human PDAC that supports CD8 T cell involvement. A trNK signature is found to be associated with better survival outcomes in PDAC and other solid tumors.

Overall, I think this is an interesting study that combines testing of therapeutic concepts in mice with bioinformatics analysis of single cell transcriptome data in primary tumors and exploration of clinical outcomes using signature genes in TCGA data. The key finding is that immunoregulatory properties of tumor infiltrating/resident CD56-bright NK cells (assumed to be non-cytotoxic) are beneficial for outcome through cross-talk with DC and recruitment of CD8 T cells. The latter is specifically induced by irradiation combined with CCR5i and PD1 blockade.

These results support the notion that IR/CCR5i/αPD1 combination treatment alters immune infiltration by reducing Tregs and increasing NK and CD8 T cells, thereby resulting in greater local tumor control.

Although the language was slightly modified in the revised version I think it is important to point out that transcripts (not protein expression) of KLRC2 is common in CD56bright NK cells and does not really reflect "adaptive-like" NK cells.

---

## [Referee Report · Reviewer #2 (Public Review)]

Summary:

This work elaborates on a combined therapeutic approach comprising ionizing radiation and CCR5i/αPD1 immunotherapy as a promising strategy in pancreatic cancer. Previous research has established that NK cell-derived CCL5 and XCL1 play a crucial role in recruiting cDC1 cells to the tumor microenvironment, contributing to tumor control. In this study, by using a murine pancreatic cancer model, the authors propose that the addition of radiation therapy to CCR5i and αPD1 immunotherapy could upregulate CD8+ T cells and a subgroup of NK cells within the tumor and result in better tumor control. They further analyzed human single-cell sequencing data from pancreatic cancer patients and identified one subgroup of NK cells (NK C1) with tissue-resident features. Subsequent cell-cell contact analysis reveals the NK-cDC1-CD8 cell axis in pancreatic cancer. By analyzing TCGA data, they found that high NK C1 signature levels were associated with better survival in pancreatic cancer patients. Thus, radiotherapy could benefit the outcome of patients bearing low NK C1 signatures. Importantly, the positive correlation between NK C1 score with survival extends beyond pancreatic cancer, showing potential applicability across various solid cancers.

Strengths:

This study could add new insight into the clinical practice by introducing such novel combined therapy and shed light on the underlying immune cell dynamics. These findings hold potential for more effective and targeted treatment in the future. Mouse experiments nicely confirmed that such combined therapy could significantly reduce tumor volume. The elegant use of single-cell sequencing analysis and human database examination enriches the narrative and strengthens the study's foundation. Additionally, the notion that NK C1 signature correlates with patient survival in various solid cancers is of high interest and relevance.

Weaknesses:

The authors have addressed some of my concerns. However, others remain and should be discussed.

(1) The role of CCR5i requires further clarification/ discussion. While the authors demonstrated its capacity to reduce Treg in murine tumors, its impact on other cell populations, including NK cells and CD8+ T cells, was not observed. Nevertheless, the effect of CCR5i on tumor growth in Figure 2B seems pathogenic. If the combination of radiotherapy and αPD1 already can achieve good outcomes as shown in Figure 3A, the necessity to include CCR5i is questioned. Overall, a more comprehensive elucidation of the roles of CCL5 and CCR5i in this context would be good. Alternatively, this limitation should be discussed.

(2) In line with this, spatial plots in Figure 4 did not include the group with only radiotherapy and αPD1. This inclusion would facilitate a clearer comparison and better highlight the essential role of CCR5i.

(3) Human database analysis showed a positive correlation between NK C1 score and CCL5 in pancreatic cancer. Furthermore, radiotherapy could benefit the outcome of patients bearing low NK C1 scores. It would be interesting to test, if radiotherapy could also benefit patients with low CCL5 levels in this cohort. This is a key question since the role of CCL5/CCR5i is not well verified. Alternatively, this point could be mentioned and discussed.

---

## [Referee Report · Reviewer #3 (Public Review)]

Summary:

In the submitted manuscript by Go et al, the authors evaluated the tumor microenvironment in pancreatic ductal adenocarcinoma (PDAC) and made a number of interesting observations, including the following: (1) CCL5 expression within the tumor microenvironment negatively correlated with clinical outcomes in human patients with PDAC; (2) there were both positive and negative correlations between CCL5 expression and the expression of specific genes (e.g. those encoding CD56 and CD16, respectively) included among gene signature lists for Treg, MDSC, TAM, and NK cells; (3) CCR5 inhibition with the inhibitor, maraviroc, reduced Treg infiltration but not that of other immune cell types in an orthotopic murine model of PDAC; (4) CCR5 inhibition augmented anti-PD1 immunotherapy when combined with ionizing radiation (IR) therapy in the murine model; (5) the above therapy resulted in increased infiltration of CD8+ cytotoxic T cells as well as of a subset of NKG2D-negative, tissue-residency (tr) marker expressing NK cells (deemed Cluster 1 NK in their data sets) that inversely correlated with the number of E-cadherin+ cells (i.e. tumor cells) and showed predicted interactions with cDC1 dendritic cells (including XCL1/XCL2 expressed by the NK and XCR1 expressed by the cDC1); (6) the authors identified a number of putative signals stemming from the trNK (e.g. IL-16, TNFSF14, FASLG, CSF, MIF) as well as incoming from cDC1s to NK (e.g. BAG6-NKp30); (7) these trNK cells positively correlated with good outcomes and with CD8+ T cell infiltrations in human PDAC as well as in many other solid tumor types; and (8) importantly, the benefit of IR therapy was specific to the subset of PDAC patients (represented in the TCGA dataset) that were predicted to have low amounts of trNK cells. The authors used murine experimental models, multi-plexed imaging analyses, and a number of publicly available sequencing data sets from human tumor samples to perform their investigations. Based on their findings, the authors proposed that combining IR with CCR5 inhibition and anti-PD1 immunotherapy is a promising strategy to treat solid cancers.

Strengths:

Overall, the collective analyses and conclusions appear to be novel and could be of high and rapid impact on the field, particularly in terms of directing clinical trials to incorporate IR with CCR5 inhibition and immunotherapy. The manuscript is well written; the figures are for the most part clear; and the Discussion is very thoughtful.

Weaknesses:

In the revised manuscript, the authors addressed my original concerns. I have no new major concerns with the study. One of the limitations is that the authors did not perform functional in vivo or ex vivo assays to address some of the major hypotheses that arose from the descriptive, correlative data; but overall, this does not detract from the enthusiasm for the work or the potential significance and impact of the study.

---

## [Author Response]

The following is the authors’ response to the previous reviews.

**Reviewer #1 (Public Review):**
Summary:The authors demonstrate that the immunosuppressive environment in pancreatic ductal adenocarcinoma (PDAC) can be mitigated by a combination of ionizing radiation (IR), CCR5 inhibition, and PD1 blockade. This combination therapy increases tissue-resident natural killer (trNK) cells that facilitate CD8 T cell activity, resulting in a reduction of E-cadherin positive tumor cells. They identify a specific "hypofunctional" NK cell population in both mouse and human PDAC that supports CD8 T cell involvement. A trNK signature is found to be associated with better survival outcomes in PDAC and other solid tumors.Strengths:Overall, I think this is an interesting study that combines testing of therapeutic concepts in mice with bioinformatics analysis of single-cell transcriptome data in primary tumors and exploration of clinical outcomes using signature genes in TCGA data. The key finding is that immunoregulatory properties of tumor-infiltrating/resident CD56-bright NK cells (assumed to be non-cytotoxic) are beneficial for outcome through cross-talk with DC and recruitment of CD8 T cells. The latter is specifically induced by irradiation combined with CCR5i and PD1 blockade."These results collectively support the notion that IR/CCR5i/αPD1 combination treatment alters immune infiltration by reducing Tregs and increasing NK and CD8 T cells, thereby resulting in greater local tumor control." I agree with this conclusion.Weaknesses:There are a few points to discuss and that the authors may want to address.(1) "Notably, CCR5i significantly reduced Treg infiltration but had no effect on the infiltration of other immune cells, indicating the active recruitment of CCR5+ Tregs in PDAC (Figure 2B)."CCR5i treatment seems to inhibit infiltration of CD8 T cells and NK cells to a greater extent, in relative terms, compared to Treg, albeit it is not statistically significant. If this visual inspection of the graph does not reflect reality, additional experiments may be needed to verify the selective targeting of Tregs or confirm the fact that also CD8 T cells and NK cells are affected by single agent CCR5i. The reduced recruitment of Treg, NK cells, and CD8T cells was completely reversed when combined with irradiation. In the data shown in Figure 3E it seems as if CCR5i induced infiltration of Tregs along with other immune cells. However, this said, I agree with the conclusion of the authors that this combined treatment leads to an altered immune composition and ratio between Tregs and effector cells (CD8T cells and NK cells). Could this altered composition be displayed more clearly?

We would like to thank the reviewer for their comments and agree that there is a trend for reduced NK and T-cell infiltration during CCR5i standalone treatment (as seen in Figure 2B), although it does not reach significance. To reflect this more clearly, we have added n.s (non-significant) for the NK cells and CD8+ T-cells and adjusted the text to reflect a trend for decreased NK and CD8+ T-cell infiltration (See Lines 162-165). Moreover, to reflect the data accurately, we have taken the Treg data out of the original Figure 2B and present it separately as a percentage of CD45+CD3+ T-cells.

(2) The definition of active and hypofunctional NK cells based on solely NKG2D expression alone seems like an oversimplification. I realize it is not trivial to test tumor-infiltrating NK cells from these tumors functionally but perhaps scRNAseq of the tumors would allow for characterization of cytotoxicity scores using KEGG or GO analysis or reversed gene set enrichment in responders/non-responders.

We agree that scRNA-seq of tumors would add to the overall characterization of the tumor-infiltrating NK cells and their characterization, however we are currently unfortunately not in the position to carry out this experiment. We did however immunophenotype the tumor infiltrating NK cell population in more depth by also looking at NKp46 and NKG2D surface expression. This newly added data demonstrates not only increased infiltration of “bona-fide” trNK cells (based on surface expression of CD103+CD49a+) under the triple treatment combination, but more importantly these trNK have reduced levels of CD69, NKp46, NKG2D and increased TIM-3 surface expression compared to conventional NK cells – suggesting that these trNKs could be more hypoactive compared to the conventional NK cells. These data have been added to the manuscript as Figure 4E, F; Figure supplement 4E-G and Lines 244-260 in the revised manuscript. To clarify this difference, we have replaced the word “hypofunctional” with “hypoactive” throughout the manuscript.

(3) It seems as if the abstract refers to this phenotype incorrectly since the "hyporesponsive" subset is described as NKG2C-negative.

We apologize for the typographic confusion and have corrected our abstract and changed the subset to NKG2D-negative (as was intended).

(4) "The NK_C1 cluster correlates best with the hypofunction NK phenotype observed in mice as similarly displayed reduced activation (reduced NKG7, NKp80, GZMA, and PRF1) with additional expression of tissue residency markers CD103, CD49a and, surprisingly, the adaptive activating receptor NKG2C (KLRC2) (Figure 5B, C)."

There is no doubt that NK_C1 represents tumor-infiltrating NK cells with a CD56bright gene signature with a strong tissue resident score. However, the transcriptional expression of KLRC2 on these is not surprising! It is well established that KLRC2 transcripts (but not protein) are highly expressed on conventional CD56bright NK cells. There are several published sources where the authors can find such data for confirmation. Thus, this is not to be confused with adaptive NK cells having an entirely different transcriptional signature and expressing high levels of NKG2C at the cell surface. I strongly recommend reinterpreting the results based on the fact that KLRC2 is expressed at high levels in conventional CD56bright NK cells. If not, it would be important to verify that these tissueresident NK cells express NKG2C and not NKG2A at the cell surface.

We agree with the reviewer and have modified the text accordingly in the revised manuscript (Lines 279-283), including references to tissue-resident adaptive-like cells as described previously in literature.

(5) NCAM1 transcript alone is not sufficient to deconvolute CD56bright NK cells in TCGA data (Figure 7A). As a single marker, it likely reflects NK cell infiltration without providing further evidence on the contribution of the bright/dim components. Therefore, the use of the bright Tr NK signature described in Table 1 is very important (Figure 7B). Table 1 is not provided. Nor Supplementary Table 1. There is only one supplementary figure in the ppt attached.

We agree that a high NCAM1/CD56 single gene signature could also represent NK cell infiltration. We have rephrased this in the text accordingly (Lines 354-357). We apologize for the missing tables and Supplementary figures. We have added these now to the manuscript as Supplementary table 1.

**Reviewer #2 (Public Review)**
Summary:This work elaborates on a combined therapeutic approach comprising ionizing radiation and CCR5i/αPD1 immunotherapy as a promising strategy in pancreatic cancer. Previous research has established that NK cell-derived CCL5 and XCL1 play a crucial role in recruiting cDC1 cells to the tumor microenvironment, contributing to tumor control. In this study, by using a murine pancreatic cancer model, the authors propose that the addition of radiation therapy to CCR5i and αPD1 immunotherapy could upregulate CD8+ T cells and a subgroup of NK cells within the tumor and result in better tumor control. They further analyzed human single-cell sequencing data from pancreatic cancer patients and identified one subgroup of NK cells (NK C1) with tissue-resident features. Subsequent cell-cell contact analysis reveals the NK-cDC1-CD8 cell axis in pancreatic cancer. By analyzing TCGA data, they found that high NK C1 signature levels were associated with better survival in pancreatic cancer patients. Thus, radiotherapy could benefit the outcome of patients bearing low NK C1 signatures. Importantly, the positive correlation between NK C1 score with survival extends beyond pancreatic cancer, showing potential applicability across various solid cancers.Strengths:This study could add new insight into the clinical practice by introducing such novel combined therapy and shed light on the underlying immune cell dynamics. These findings hold potential for more effective and targeted treatment in the future. Mouse experiments nicely confirmed that such combined therapy could significantly reduce tumor volume. The elegant use of single-cell sequencing analysis and human database examination enriches the narrative and strengthens the study's foundation. Additionally, the notion that NK C1 signature correlates with patient survival in various solid cancers is of high interest and relevance.Weaknesses:The role of CCR5i requires further clarification. While the authors demonstrated its capacity to reduce Treg in murine tumors, its impact on other cell populations, including NK cells and CD8+ T cells, was not observed. Nevertheless, the effect of CCR5i on tumor growth in Figure 2B should be shown. If the combination of radiotherapy and αPD1 already can achieve good outcomes as shown in Figure 3A, the necessity to include CCR5i is questioned. Overall, a more comprehensive elucidation of the roles of CCL5 and CCR5i in this context would be good.

We would like to thank the reviewer for their comments and agree that standalone CCR5i also shows a trend of reduced infiltrating NK cells and CD8+ T-cells, although this does not reach significance. We have mentioned this trend in the manuscript (see Lines 162-165) and added n.s to Figure 2B as well. In regards to adding CCR5i; although we observe volumetric control by radiotherapy and anti-PD1, we observe an increase in necrosis induction only in the triple combination compared to radiotherapy combined with anti-PD1 – suggesting that there is an additive effect of CCR5i in our model only as a combination modality. We therefore believe that addition of CCR5i to radiotherapy and anti-PD1 has a beneficial effect. The growth curves for CCR5i alone were already presented in Figure 3A, and we have modified our manuscript to refer to this (see Lines 165-167).

(1) In line with this, spatial plots in Figure 4 did not include the group with only radiotherapy and αPD1. This inclusion would facilitate a clearer comparison and better highlight the essential role of CCR5i.

We agree with the reviewer that inclusion of radiotherapy and αPD1 would facilitate a clear comparison of our data and our experiments did include single controls for radiotherapy and αPD1; however, unfortunately, the tissue slides were of bad quality and therefore not suitable for quantification. In line with this, we have added references to other studies that investigated the effect of immune checkpoint inhibitors in combination with radiotherapy (see Lines 169-172).

(2) NK C1 cells should be also analyzed in the mouse model. The authors suggest that NKNKG2Dve could be the cell population. Staining of inhibitory markers should be considered, for example, TIGIT and TIM3 as presented in Figure 5B.

As per the reviewer suggestion, we have now included some additional data on the surface expression of inhibitory markers/activating receptor on tumor-infiltrating NK cells in our model under the triple combination. These additional data demonstrate increased infiltration of trNK under the triple combination that seem to be more ‘hypoactive’ than conventional NK cells. This data has been added as Figure 4E in the revised Figure.

(3) While the cell-cell contact analysis generated from single-cell sequencing data is insightful, extending this analysis to the mouse model under therapy would be highly informative. NK and CD8 cells in the tumor increased upon the combined therapy. However, cDC1 was not characterized. Analysis regarding cDC1 would provide more information on the NK/cDC1/CD8 axis.

We agree that looking into cDC1 would be highly interesting in our treatment model and its characterization is currently under investigation. The importance about the interaction between cDC1-NK cells has been described before by various groups, and we have provided additional references for that in our manuscript (see Lines 449-455)

(4) Human database analysis showed a positive correlation between NK C1 score and CCL5 in pancreatic cancer. Furthermore, radiotherapy could benefit the outcome of patients bearing low NK C1 scores. It would be interesting to test if radiotherapy could also benefit patients with low CCL5 levels in this cohort.

We would like to thank the reviewer for their suggestion and please see the figure below for the comparison. Patients with CCL5high are enriched for NK_C1 (Figure 7D) and CCL5high patients with NK_C1high have significantly increased overall and disease-free survival compared to NK_C1low (Figure 7E); where those with NK_C1low significantly benefit from radiotherapy (Figure 7B). Accordingly, patients with CCL5high have significantly decreased overall survival compared to CCL5low patients, again confirming CCL5 as a prognostic marker (Figure 1A, Figure R1). When we look at CCL5low patients however, there is no additional significant benefit for radiotherapy (see insert below) in the CCL5low group (not significant; only significant p-values are shown). These data collectively support the strong correlation between CCL5 levels and NK_C1 enrichment, and imply that radiotherapy alone is insufficient to drive NK_C1 cells in the absence of high CCL5 gradients to improve overall survival. However, given the increased overall survival of CCL5low compared to CCL5high it is likely that other factors are at play. Future studies will be required to further elucidate the role of CCL5 gradients on NK_C1 cells and the beneficial effect of radiotherapy.

**Author response image 1. sa4fig1:** Overall survival of CCL5high versus CCL5low patients stratified into groups with and without radiotherapy using TCGA-PAAD. Log-rank p-value indicates the significance level across all groups while individual significant comparisons are shown as indicated.

**Reviewer #3 (Public Review):**
SummaryIn the submitted manuscript by Go et al, the authors evaluated the tumor microenvironment in pancreatic ductal adenocarcinoma (PDAC) and made a number of interesting observations, including the following: (1) CCL5 expression within the tumor microenvironment negatively correlated with clinical outcomes in human patients with PDAC; (2) there were both positive and negative correlations between CCL5 expression and the expression of specific genes (e.g. those encoding CD56 and CD16, respectively) included among gene signature lists for Treg, MDSC, TAM, and NK cells; (3) CCR5 inhibition with the inhibitor, maraviroc, reduced Treg infiltration but not that of other immune cell types in an orthotopic murine model of PDAC; (4) CCR5 inhibition augmented anti-PD1 immunotherapy when combined with ionizing radiation (IR) therapy in the murine model; (5) the above therapy resulted in increased infiltration of CD8+ cytotoxic T cells as well as of a subset of NKG2D-negative, tissueresidency (tr) marker expressing NK cells (deemed Cluster 1 NK in their data sets) that inversely correlated with the number of E-cadherin+ cells (i.e. tumor cells) and showed predicted interactions with cDC1 dendritic cells (including XCL1/XCL2 expressed by the NK and XCR1 expressed by the cDC1); (6) the authors identified a number of putative signals stemming from the trNK (e.g. IL-16, TNFSF14, FASLG, CSF, MIF) as well as incoming from cDC1s to NK (e.g. BAG6-NKp30); (7) these trNK cells positively correlated with good outcomes and with CD8+ T cell infiltrations in human PDAC as well as in many other solid tumor types; and (8) importantly, the benefit of IR therapy was specific to the subset of PDAC patients (represented in the TCGA dataset [55]) that were predicted to have low amounts of trNK cells. The authors used murine experimental models, multiplexed imaging analyses, and a number of publicly available sequencing data sets from human tumor samples to perform their investigations. Based on their findings, the authors proposed that combining IR with CCR5 inhibition and anti-PD1 immunotherapy is a promising strategy to treat solid cancers.StrengthsOverall, the collective analyses and conclusions appear to be novel and could be of high and rapid impact on the field, particularly in terms of directing clinical trials to incorporate IR with CCR5 inhibition and immunotherapy. The manuscript is well written; the figures are for the most part clear; and the Discussion is very thoughtful.WeaknessesThere were a number of minor typographical errors, missing references, or minor issues with the figures. In general, while many of the observations provided strong suggestive evidence of relationships, phenotypes, and functions, the authors often used language to indicate that such things were confirmed, validated, or proven. In fact, there was a paucity of such functional/confirmatory experiments. This does not necessarily detract from the overall significance, excitement for, and potential impact of the study; but the language could likely be adjusted to be more in keeping with the true nature of the findings. The main title and running title are a bit different; consider making them more similar.

We apologize for the typographical errors, missing references and issues with the figures. We have revised our manuscript, with a major focus on adjusting our language to more carefully reflect our data, and hope to have addressed all the concerns of the reviewer. The slight discrepancy between the main title and running title are to be able to convey the contents of this manuscript in a comprehensive way.

**Recommendations for the authors:**

**Reviewer #1 (Recommendations For The Authors):**
Please make sure all files are made available. Also please check available datasets describing KLRC2 transcripts in CD56brights. This is not to be confused with an adaptive-like signature.

We have added the missing table to the supplementary figures and revised the manuscript text in regards to KLRC2 transcript in our NK_C1 cluster and its implications for an adaptive-like signature in the context of tissue-residency (see Lines 279-283; 465-474).

**Reviewer #2 (Recommendations For The Authors):**
Additional experiments as mentioned in the 'weakness' section could help to further strengthen this study. Besides these points, I would recommend the following:(1) The description in the figure should be more precise and clear. Especially in Figure 3A, it seems the addition of IR into CCR5i or CCR5i/aPD1 leads to a bigger tumor volume.

We have adjusted the figure descriptions to more clearly describe the figures. We apologise for the confusion in Figure 3A, this was a figure legend error and has been correctly rectified in the revised Figures (i.e. closed symbols represent +IR conditions).

(2) The definition of Tregs in figures should be described, e.g. it is not specified which population is shown in Figure S2c.

We have added a definition of Tregs (i.e. Live/CD45+CD3+CD4+FOXP3+) in our revised manuscript (see Lines 162-165). To avoid confusion, we have removed the subsequent gating of CCR5 and PD-1 of Tregs in our revised Supplementary Figures.

(3) Please add a bar in all histology figures, for example, Figure 2A, S2A, S3E. It seems in Figure S3D, E, the green group is missing.

We have added the scale bar to all the indicated figures. Unfortunately, indeed as correctly pointed out by the reviewer, we are missing the green group (i.e. IR+CCR5i) as we felt that the excessive growth seen with CCR5i alone may have given a false impression of the extent of infiltration, therefore we did not include this in the original analysis and do not have the data in the Figure.

(4) Please check through the manuscript, there are some grammar mistakes.

We apologise for the grammar mistakes in our original manuscript and have carefully revised the current manuscript to avoid grammar mistakes

(5) Figure S7B, the left cell lacks a name.

We have annotated the left cell accordingly in our revised supplementary figure.

**Reviewer #3 (Recommendations For The Authors):**
(1) Abbreviations (e.g. PDAC) should be spelled out the first time introduced in the manuscript.

We have adjusted this in our revised manuscript.

(2) Referring to the tissue-resident NK cells as "hypofunctional" may not be useful...they seem to be functional, just not in the conventional sense. The authors may want to consider another term, such as non-cytotoxic (given the low expression of cytolytic granules, etc) or immunoregulatory (as they actually refer to them on line 310).

We agree with the reviewer and have revised the manuscript to refer to them as “immunoregulatory” or “hypoactive” when appropriate. The latter is supported by the additional experiments as shown in Figure 4E.

(3) Barry et al 2018 Nat Med demonstrated that NK cells in melanoma could support cDC1s and promote positive clinical outcomes in the setting of immunotherapy. It would likely be beneficial to also cite this paper (e.g. on line 425).

Thank you for the suggestion, which would work in line with our hypothesis of crosstalk between NK_C1 and cDC1. We have looked for FLT3L in our NK_C1 cluster and did not find any enrichment for FLT3L transcript (see Figure 5E). Nevertheless, we have added the reference in the discussion of our manuscript to further support the importance of crosstalk between cDC1 and NK cells (see Lines 449455)

(4) Figure 2B: by eye, it looks like the difference between CD8+ T cells in the two conditions would be significantly different; is this not the case? Same thing for the NK cells...what are the pvalues?

We have added n.s. to our revised Figure 2B. The p-values for CD8+ T-cells and NK cells were 0.14 and 0.19 {2-tailed students t-test}, respectively.

(5) The murine data strongly suggest that the combination therapy promotes trNK cell infiltration into the tumors, in turn resulting in cDC1-mediated CD8+ T cell infiltration and/or activation. It could be highly valuable/useful to functionally determine (e.g. by depleting NK cells in this model) if NK cells are required for the effects seen.

We agree that depletion of NK cells could really solidify the findings even more, and it is part of ongoing investigations for future projects. However, it would be imperative to first characterise these NK cells in more depth as conventional global ablation of NK cells is excepted to highly impact immunosurveillance as well. This is part of current ongoing work.

(6) Figure 7B: how were "high" and "low" defined (for the NK signature)?

An enrichment score of the NK_C1 gene signature (see Table supplement 1) was first calculated per patient sample in the TCGA RNA-seq dataset using the Gene Set Variation Analysis (GSVA) method. A cut-off value was then determined using the maximally selected rank statistics (max-stat R package) method to divide patients into “high” and “low”.

(7) Lines 164-165 of the Results: it would be good to include a reference supporting the statement.

We have added rephrased the manuscript and added corresponding references (see Lines 170-173 in revised manuscript).

(8) There are many conclusions and very speculative language based only on sequencing results, and these have not been validated (e.g. in the Discussion, lines 447-453). As another example, it was concluded that a decrease in NKG2D+ NK cells implied a reduction in overall NK cell cytolytic activity and that NKG2D- NK cells were hypofunctional and did not kill well. This was not tested. Generally, it would be useful for the authors to use language that conveys that the data are primarily suggestive (rather than "confirmatory", line 447) of relationships, phenotypes, and functions at this point.

We thank the reviewer for their concerns and have carefully adapted the manuscript text to more clearly clarify the findings in a careful manner.

(9) On lines 246-247 the authors refer to cluster 3 NK cells, which express CD16, as "immature". The rationale for this designation is not provided, and most human NK cell development models hold that CD16+ NK cells represent the most mature subset(s).

We apologize for the typographic error – later on we refer to the NK_C3 cluster as cytotoxic NK cells and we have corrected this in our revised manuscript (see Lines 273-275).

(10) On line 351, the authors reference supplemental Figure 7C...but I don't see this figure in the accompanying powerpoint file.

This should have been Supplementary Figure 7B, and we have corrected it in the revised manuscript (see Lines 374-377)

(11) On line 417, the authors reference NKp40; this is likely a typographical error.

This has been corrected in the revised manuscript to NKp46 (see Lines 439-442).